# Electroacupuncture improves swallowing function in a post-stroke dysphagia mouse model by activating the motor cortex inputs to the nucleus tractus solitarii through the parabrachial nuclei

Lulu Yao[1,5], Qiuping Ye[1,2,5], Yun Liu[1,3,5], Shuqi Yao[1,5], Si Yuan[1,5], Qin Xu[1], Bing Deng[1], Xiaorong Tang[1], Jiahui Shi[1], Jianyu Luo[1], Junshang Wu[1], Zhennan Wu[1], Jianhua Liu[1,4], Chunzhi Tang[1], Lin Wang[1] ✉ & Nenggui Xu[1] ✉

As a traditional medical therapy, stimulation at the *Lianquan* (CV23) acupoint, located at the depression superior to the hyoid bone, has been shown to be beneficial in dysphagia. However, little is known about the neurological mechanism by which this peripheral stimulation approach treats for dysphagia. Here, we first identified a cluster of excitatory neurons in layer 5 (L5) of the primary motor cortex (M1) that can regulate swallowing function in male mice by modulating mylohyoid activity. Moreover, we found that focal ischemia in the M1 mimicked the post-stroke dysphagia (PSD) pathology, as indicated by impaired water consumption and electromyographic responses in the mylohyoid. This dysfunction could be rescued by electroacupuncture (EA) stimulation at the CV23 acupoint (EA-CV23) in a manner dependent on the excitatory neurons in the contralateral M1 L5. Furthermore, neuronal activation in both the parabrachial nuclei (PBN) and nucleus tractus solitarii (NTS), which was modulated by the M1, was required for the ability of EA-CV23 treatment to improve swallowing function in male PSD model mice. Together, these results uncover the importance of the M1-PBN-NTS neural circuit in driving the protective effect of EA-CV23 against swallowing dysfunction and thus reveal a potential strategy for dysphagia intervention.

Annually, approximately 15 million people worldwide suffer from a stroke[1]. Of these individuals, as many as 78% develop post-stroke dysphagia (PSD, Supplementary Table 1), defined as difficulty in swallowing after stroke[2]. In addition to malnutrition, over 20% of PSD patients develop aspiration pneumonia, which is the leading cause of death after discharge[3]. Current dysphagia treatments, which have been developed over the past decades, include chemical, physical, and electrical methods for stimulating the peripheral oropharyngeal sensory system and exciting peripheral nerve fibers from the pharynx[4,5]. Among them, pharyngeal electrical stimulation (PES) and laryngopharyngeal neuromuscular electrical stimulation (NMES), as peripheral stimulation methods, are commonly used to treat dysphagia[6–8]. Moreover, acupoint stimulation, a type of traditional superficial peripheral stimulation approach, has been utilized in clinical practice for

the treatment of diseases and various conditions, especially pain, for more than 2500 years[9,10]. Regarding stroke rehabilitation, acupuncture treatment has been widely used to promote the recovery of both sensory and motor functions[11].

*Lianquan* (CV23), an acupoint located in the depression superior to the hyoid bone and on the anterior median line[12] (Fig.1a), has been widely selected to treat swallowing disorders, especially PSD[13–15]. In clinical practice, several assessments have been adopted to evaluate swallowing function following stimulation at CV23, including the Video fluoroscopic swallowing study (VFSS, measures the pharyngeal transit time, esophageal transit time and size of food), Fiberoptic endoscopic evaluation of swallowing (FEES, assesses the structure of pharynx and larynx), Standardized swallowing assessment (SSA, measures the consciousness, body control, breathing, oral closure, laryngeal function, pharyngeal reflex and spontaneous cough, etc.), the Kubota water swallowing test (WST, evaluates the presence of cough/wet voice after

swallowing), and Pharyngeal surface electromyography (PsEMG, evaluates of the characteristics of the external musculature by quantifying the root mean square, integrated EMG and averaged EMG)[16–20]. Although CV23 acupoint stimulation is widely used, the mechanism by which it ameliorates PSD remains unknown. In this study, we aimed to uncover the mechanism through which electroacupuncture at the CV23 acupoint (EA-CV23) improves swallowing function in a PSD mouse model.

Previous studies have suggested that the onset of voluntary swallowing activities requires cortical function[21–24], and dysfunction of the primary motor cortex (M1) might be associated with the occurrence of dysphagia[25–27]. Some researchers have proposed that recovery of swallowing function in PSD patients is related to increased excitability of the pharyngeal motor cortex in the contralateral hemisphere[28–32]. Although these results suggest the importance of the M1 in the regulation of swallowing activity, the localization and types

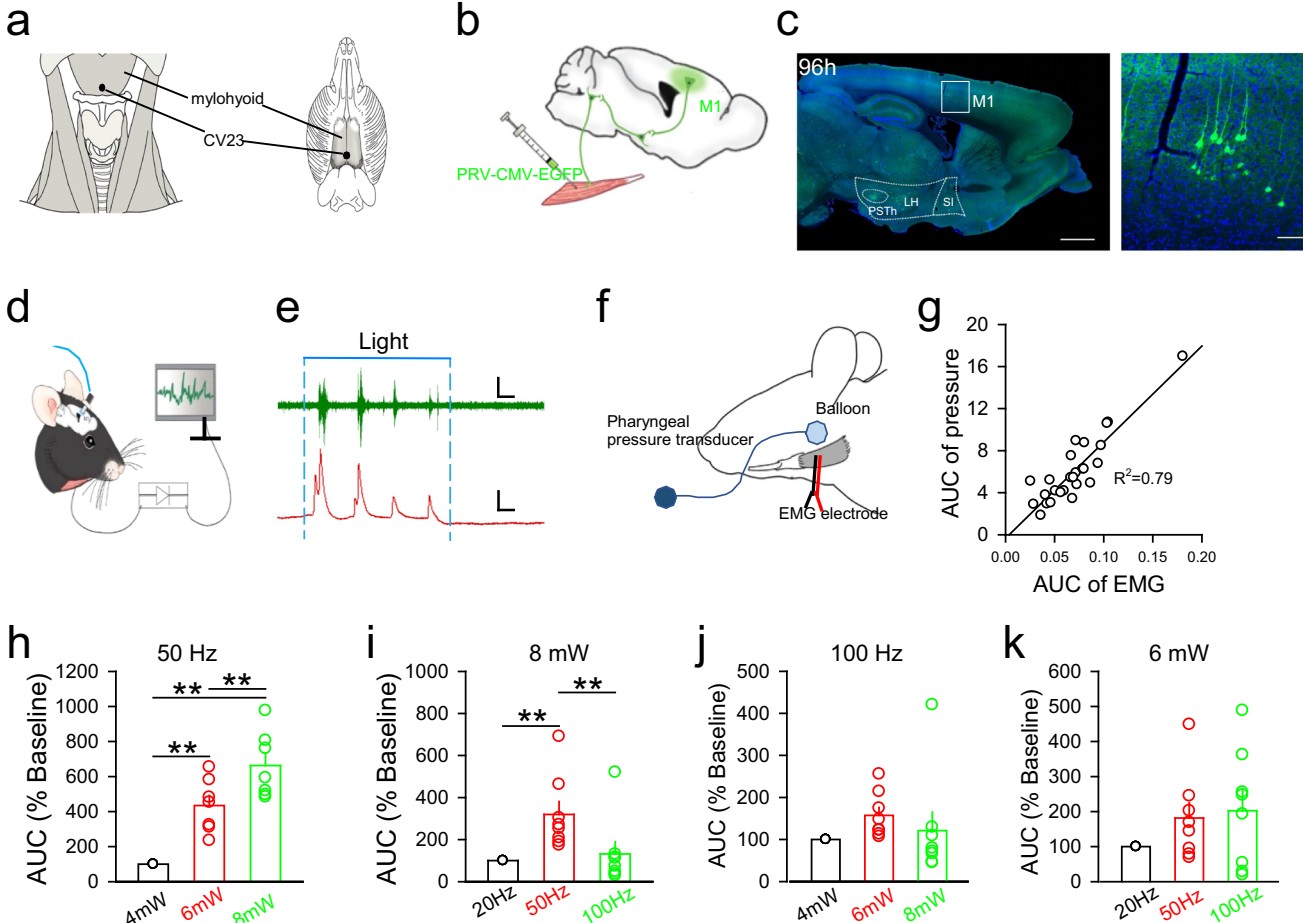

**Fig. 1 | Optogenetic activation of M1 L5 neurons controlled mylohyoid activity.**
**a** The position of CV23 acupoint in the coronal plane in humans (left) and mice (right). **b** Schematic diagram of the experiment for the retrograde tracing of PRV in adult mice. PRV was injected into the mylohyoid, and EGFP-labeled neurons were observed in M1. **c** Left: the distribution of PRV-infected neurons (green) in the M1 at 96 h after injection. DAPI (blue) was used for counterstaining, Scale bar, 1000 μm. Right: enlarged version of the boxed image on the left. Scale bar, 100 μm.
**d**, **e** Schematic diagram of experiment for EMG recordings from the mylohyoid following optogenetic activation of ChR2-expressing excitatory neurons in the M1 by a blue laser (473 nm) (**d**). Representative traces of EMG responses (green) and the pharyngeal pressure (red) induced by light stimulation. The blue solid line indicates the duration of light stimulation (5 s), and the blue dashed line indicates the start and the end of light stimulation. Scale bar, top: 1 s and 2 mV; bottom: 1 s and 5 mmHg (**e**). **f** The experimental scheme for the detection of pharyngeal

pressure during EMG recording. **g** The relationship between the AUC of EMG responses and AUC of pharyngeal pressure induced by optogenetic activation of the M1. Each point represents a swallowing process. **h** The AUC of EMG responses 1 at different powers (4, 6, and 8 mW) at 50 Hz of blue light stimulation at M1. (one-way ANOVA, $N = 8$ per group; $F = 28.57$, **$P < 0.01$). **i** The AUC of EMG responses after 8 mW blue light stimulation of the M1 at different frequencies (20 Hz, 50 Hz, and 100 Hz) (one-way ANOVA, $N = 8$ per group; $F = 5.905$, **$P < 0.01$, two-side analysis with post hoc Tukey's test). **j** The EMG responses at different intensities (4, 6, and 8 mW) at 100 Hz of blue laser given in M1 (one-way ANOVA, $N = 8$ per group; $F = 1.095$, $P = 0.2088$). **k** The EMG responses at different frequencies (20, 50, and 100 Hz) at 6 mW of blue laser given in M1 (one-way ANOVA, $N = 8$ per group; $F = 1.551$, $P = 0.2354$, two-side analysis with post hoc Tukey's test). Data are presented as Mean ± SEM. $N$ indicates the number of biologically independent samples, mice per group.

of M1 neurons involved in this process remain elusive. The nucleus tractus solitarii (NTS), located in the brainstem, is one of regions of swallowing-related central pattern generators (CPGs) and well-recognized to be important for the swallowing reflex. Evidence showed that the NTS can be modulated by the M1[33,34]. In addition, projections from laryngeal and pharyngeal motor cortical regions to the parabrachial nucleus (PBN) have been identified[33,35–37], and neurons in the PBN contribute to taste sensation and feeding behavior, suggesting the involvement of the PBN in swallowing[38]. These findings collectively suggest that the M1, PBN, and NTS may interact to regulate the swallowing process. However, there is no direct evidence illustrating how these brain regions jointly participate in the regulation of swallowing function.

In this study, we found that selective activation of M1 layer 5 (L5) excitatory neurons using optogenetics, a method with exquisite spatial and temporal precision[39], elicited frequency- and intensity-dependent EMG responses in the mylohyoid. In addition, unilateral photochemical ischemia in the M1 could induce dysphagia pathology, as indicated by impairment of water consumption and EMG responses in the mylohyoid, and thus was used to establish a PSD mouse model. Furthermore, the activity of excitatory neurons in the M1 was required for the ability of EA-CV23 treatment to improve swallowing function in PSD model mice. Subsequently, we demonstrated the interaction of the M1 with subcortical brain regions, including PBN and NTS, and addressed the importance of the M1-PBN-NTS neural circuit in driving the swallowing process and the therapeutic effect of EA-CV23 treatment against PSD.

## Results

### Optogenetic activation of M1 L5 neurons controls mylohyoid activity

The swallowing process requires simultaneous activation of a cluster of pharyngeal muscles, including the mylohyoid where the CV23 acupoint is located (Fig. 1a, Supplementary Fig. 1a, b)[40]. To validate whether EA-CV23 improves swallowing function via the M1-mylohyoid pathway, we first determined whether there are synaptic connections between M1 neurons and the mylohyoid. A transsynaptic retrograde pseudorabies virus (PRV; PRV-CMV-EGFP) was injected into the mylohyoid (Fig. 1b, Supplementary Table 2), and the EGPF-positive neurons were observed in M1 L5 at 96 h but not 48 h or 72 h post-injection (Fig. 1c, Supplementary Fig. 1c). In general, the infected animals could survive for 3-5d after PRV injection[41]. The results indicated that there might have been more than 3 transsynaptic connections through which the PRV was transported from the mylohyoid to M1.

Then we investigated whether the function of these connections was associated with the regulation of swallowing activity. The AAV2/9-CaMKIIα-ChR2-mCherry virus, which is widely used in combination with blue light stimulation (473 nm) to optogenetically activate neurons[42], was injected into M1 L5, and EMG responses were recorded in the mylohyoid to assess the mylohyoid activity. 21 days after injection, the expression of the virus was confirmed by the presence of mCherry-positive labeling neurons (Supplementary Fig. 1d, e), while the function of virus was ascertained by in vitro slice electrophysiological recording of membrane potential depolarization and action potential firing induced by blue light stimulation (Supplementary Fig. 1f, g). Optogenetic activation of M1 neurons induced an increase in the mylohyoid activity and pharyngeal pressure (Fig. 1d, e), the latter of which was measured by a balloon filled with water and held in the pharyngeal position near the root of the tongue. The area under the curve (AUC) of EMG responses in the mylohyoid was well correlated with the pharyngeal pressure (Fig. 1f, g), suggesting that the reliability of EMG responses in the mylohyoid for reflecting swallowing function. The optogenetic stimulation parameters that elicited the maximum EMG responses (8 mW, 50 Hz) were determined (Fig. 1h–k) ($P < 0.01$). Together, these results demonstrated

that M1 L5 neurons regulated swallowing function by modulating mylohyoid activity.

### PSD pathology can be mimicked by photothrombotic ischemia in the M1 and attenuated by EA-CV23

To investigate whether swallowing difficulty can be induced by M1 injury, we first induced focal ischemia in the M1 of mice by intraperitoneal injection of Rose Bengal (RB) and laser irradiation of the M1 region (Fig. 2a). Blood flow was decreased in the ischemic M1 and prefrontal cortex (PFC) compared to that in the contralateral regions, but there was no difference in blood flow between the contralateral and ipsilateral M1 among these non-ischemia groups (Fig. 2b, c) ($P < 0.01$). Compared with mice in the non-ischemia and PFC ischemia groups, mice in the M1 ischemia group exhibited swallowing difficulty, as shown by the impaired AUC of EMG responses in the mylohyoid induced by water delivery (Fig. 2d, e, Supplementary Fig. 2a) ($P < 0.01$), as well as the decreased water consumption in mice with M1 ischemia (Fig. 2f) ($P < 0.01$). These results suggested that dysphagia could be induced by photothrombotic ischemia in the M1, and thus proved the successful establishment of the PSD mouse model. These observations were consistent with our previous results[43–45], which indicated that focal ischemia in the M1 can induce impaired EMG responses in the mylohyoid and decreased water consumption in mice. Next, to determine whether EA-CV23 attenuated the swallowing difficulty in PSD model mice, EMG recordings in the mylohyoid and the water consumption test were performed 24 h following EA-CV23. In addition to the AUC, all the amplitude, frequency, and the onset latency of EMG responses were significantly increased in PSD mice after EA-CV23 (Fig. 2g–i, Supplementary Fig. 2b–d) ($P < 0.01$). Collectively, these results indicated that EA-CV23 ameliorated swallowing dysfunction induced by focal ischemia in the M1.

### The effect of EA-CV23 on PSD depends on the activity of excitatory neurons in the contralateral M1 L5

We next determined whether EA-CV23 improved swallowing function by modulating the neuronal activity in contralateral M1 L5. The expression of c-Fos (a marker of cell activation) in the CaMKIIα (a marker of excitatory neurons)- or the GAD67 (a marker of inhibitory neurons)- positive neurons in the M1 was examined by immunofluorescence staining, and the results showed that the excitatory neurons in M1 L5 were activated by EA-CV23 in PSD model mice (Fig. 3a, b, Supplementary Fig. 3) ($P < 0.05$, $P < 0.01$). Then, to assess EA-CV23-induced activation of contralateral M1 L5 excitatory neurons in vivo, AAV2/9-CaMKIIα-GCaMP6s virus, as a calcium ($Ca^{2+}$) indicator whose fluorescence intensity reflects the activity of excitatory neurons, was injected into M1 L5. After 21 d, two-photon laser microscopy imaging showed that individual somatic $Ca^{2+}$ transients in M1 L5 were increased upon EA-CV23 in PSD model mice (Fig. 3c–e) ($P < 0.01$). While fiber photometry recording revealed that population neuronal activity in M1 L5 was also potentiated by EA-CV23, and this potentiation was much greater than that observed in the PFC, and Sham EA and EA at another acupoint (PC6) had little effect on the population neuronal activity in M1 L5 (Supplementary Fig. 4). Together, both the in vitro and in vivo results suggested that EA-CV23 activated the contralateral M1 L5 excitatory neurons in PSD model mice.

Next, we sought to test whether the activity of contralateral M1 L5 excitatory neurons was required for the effect of EA-CV23 in PSD model mice. To this end, AAV2/9-CaMKIIα-hM4Di-mCherry, which is widely used in combination with Clozapine-N-oxide (CNO) injection (3 mg/kg) to chemogenetically inhibit neurons[46], was injected into M1 L5. The EA-CV23-induced potentiation of EMG responses in the mylohyoid and increase in water consumption, and the expression of c-Fos in PSD mice were blocked after inhibiting M1 L5 excitatory neurons

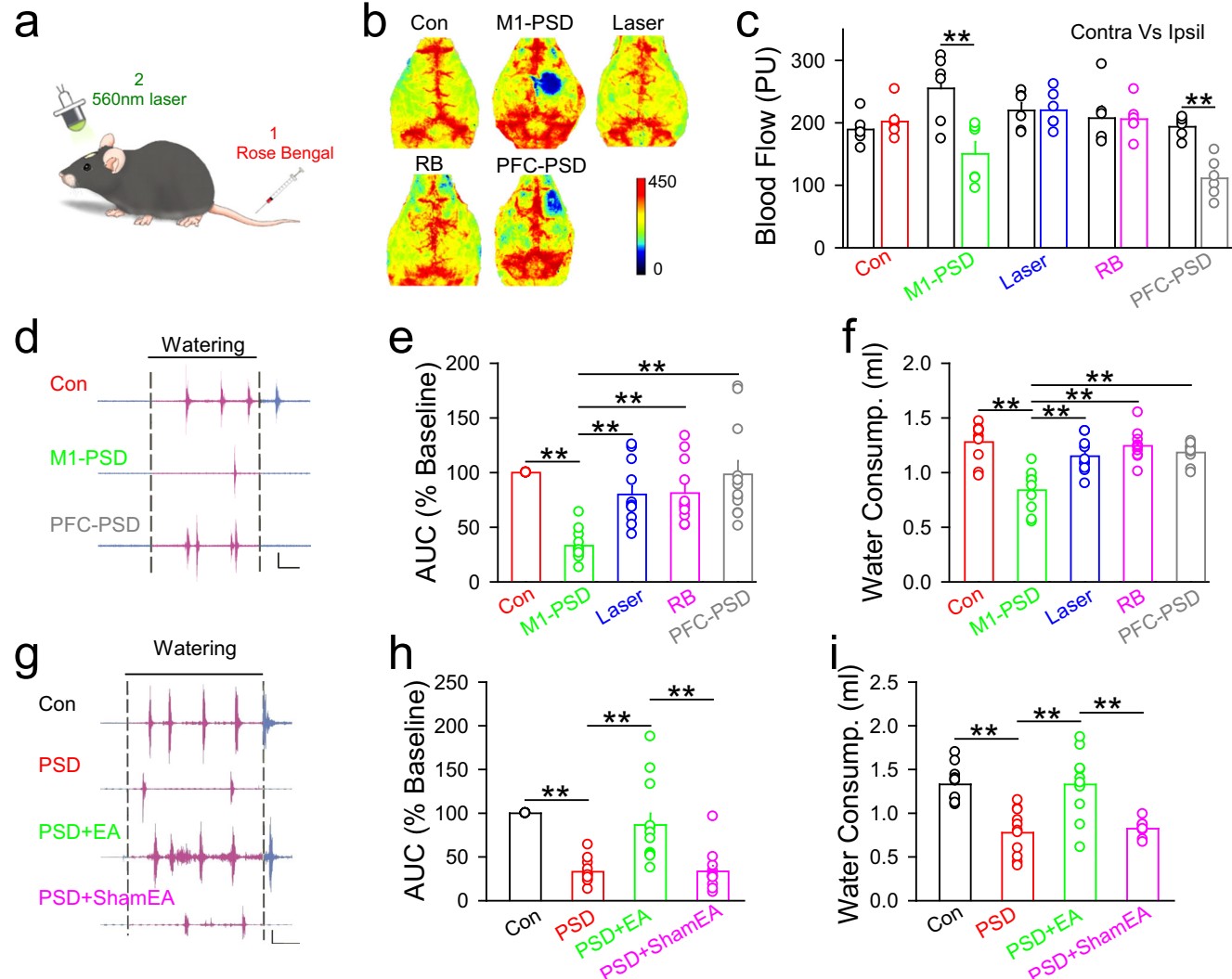

**Fig. 2 | EA-CV23 improved swallowing function in a mouse model of dysphagia induced by unilateral photothrombotic ischemia in the M1. a** The strategy used to induce ischemia in mice. RB (15 mg/ml, RB) was intraperitoneally injected, and then a 560 nm laser was used to focally stimulate the M1. **b** Representative laser speckle contrast images of thrombotic infarction in the different groups, including the control (Con), M1 ischemia (M1-PSD), laser illumination only (Laser), RB only and PFC ischemia (PFC-PSD) groups. **c** Blood perfusion in the contralateral (Contra) and ipsilateral (Ipsil) brain hemispheres after treatment in the different groups (one-way ANOVA, $N = 6$ per group; $F = 7.529$, **$P < 0.01$). **d, e** Representative traces of water-induced EMG responses in the different groups, including the Con, M1-PSD, and PFC-PSD groups. Black solid line: duration of water delivery. Black dashed lines: the start and end of water delivery. Scale bar, 2 s and 0.5 mV (**d**). Water-induced EMG responses decreased significantly in M1-PSD group, compared to the Con, Laser, RB, and PFC-PSD groups (one-way ANOVA, $N = 10$ per group; $F = 12.34$, **$P < 0.01$) (**e**). **f** Water consumption was decreased in the M1-PSD group, compared to the Con, Laser, RB, and PFC-PSD groups (one-way ANOVA, $N = 10$ per group; $F = 12.76$, **$P < 0.01$). **g, h** Representative traces of water-induced EMG responses in the control (Con), PSD model mice (PSD), EA-treated PSD model mice (PSD + EA), and sham EA-treated PSD model mice (PSD + ShamEA) groups (black solid line: duration of water delivery; black dashed lines: start and the end of water delivery). Scale bar, 2 s and 0.5 mV (**g**). The AUC of EMG responses induced by water delivery was decreased in PSD model mice, and this change could be attenuated by EA treatment (**h**) (one-way ANOVA, $N = 12$ per group; $F = 20.63$, **$P < 0.01$). **i** The decreased water consumption could be restored by EA treatment (one-way ANOVA, $N = 11$ per group; $F = 16.52$, **$P < 0.01$). Data are presented as Mean ± SEM. $N$ indicates the number of biologically independent samples, mice per group.

(Fig. 3f–h, Supplementary Fig. 5) ($P < 0.01$). Overall, these results supported the essential contribution of contralateral M1 L5 excitatory neurons to the improvement of swallowing function elicited by EA-CV23 in PSD model mice.

**The activity of PBN neurons is required for the effect of EA-CV23 on swallowing dysfunction in PSD model mice and is regulated by M1 activation**
EA-CV23 was observed to increase expression of c-Fos in the PBN, as well as the M1, NTS, paraventricular hypothalamus (PVH), hypothalamus, and thalamus, in normal mice (Fig. 4a, b, Supplementary Fig. 6) ($P < 0.01$). This increase in c-Fos expression was also detected in the PBN of PSD model mice after EA-CV23 (Fig. 4c, d) ($P < 0.05$).

These results indicated that neurons in the PBN could be activated by EA-CV23 under both physiological and pathological conditions. The role of PBN neurons in the effect of EA-CV23 on swallowing function was further confirmed by inhibiting PBN neurons via injection of AAV2/9-CaMKIIα-hM4Di-mCherry virus into the PBN and subsequent intraperitoneal injection of CNO. We found that the EA-CV23-mediated of improvement of EMG responses induced by water delivery was abolished after inhibition of PBN neurons and that inhibition of these neurons also decreased the EMG responses in the mylohyoid under both physiological conditions (Fig. 4e, f) ($P < 0.01$), and in PSD (Fig. 4f–h) ($P < 0.01$). Together, these results clarified the crucial role of PBN neurons in swallowing function and the ability of EA-CV23 to treat PSD.

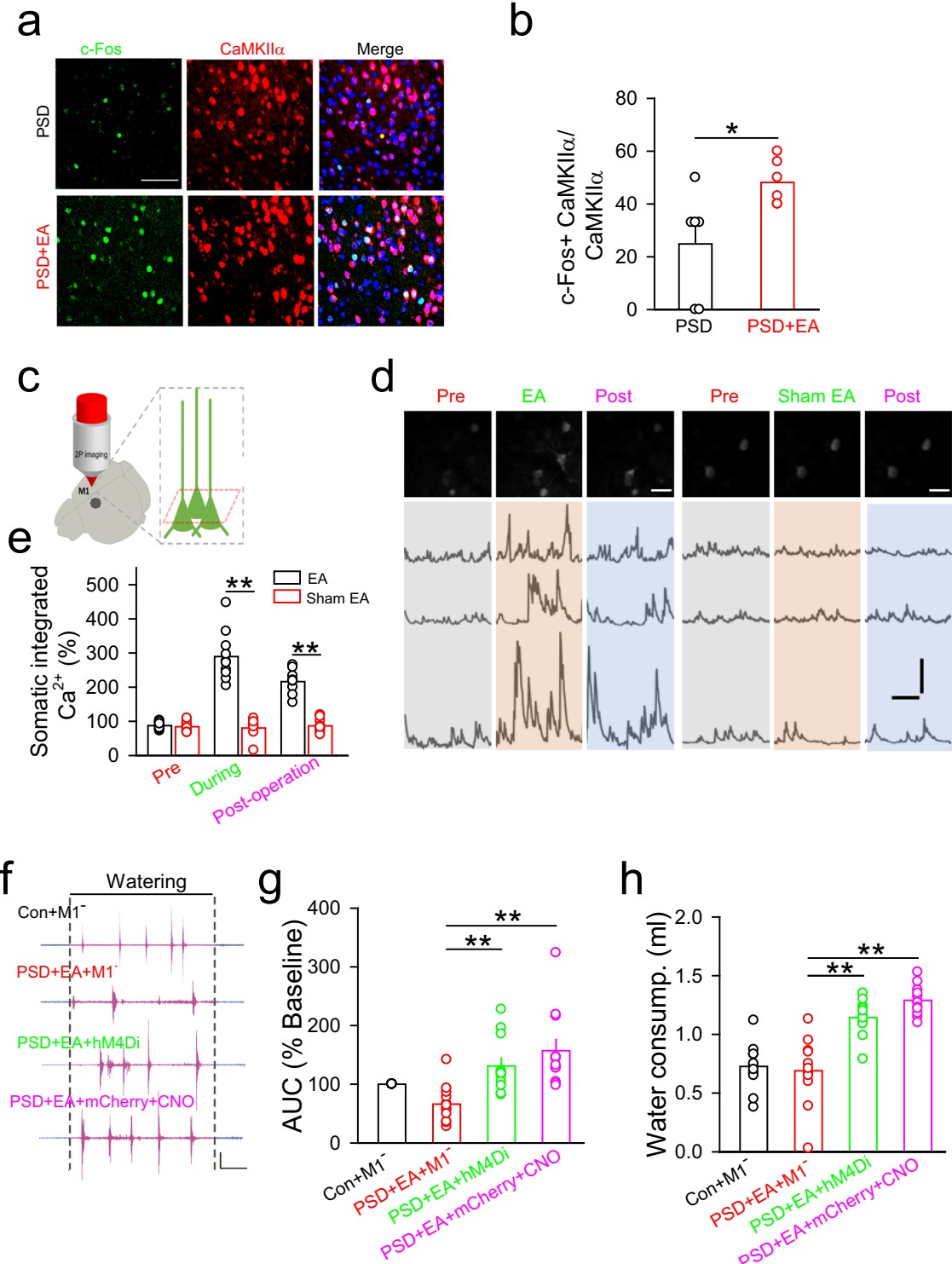

We next validated whether PBN neurons could interact with M1 neurons and thus contribute to the EA-CV23-mediated improvement of swallowing function. Optogenetic activation of M1 L5 neurons increased the expression of c-Fos in the PBN (Fig. 5a, b) (P < 0.01); in contrast, inhibition of PBN neurons decreased mylohyoid activity induced by optogenetic activation of the M1 (Fig. 5c, d) (P < 0.01), suggesting that the PBN is involved in M1-regulated swallowing function. Then, we further selectively activated the terminals of neurons projecting from the M1 to the PBN by injecting AAV2/9-CaMKIIα-ChR2-mCherry virus into M1 and implanting a fiber into the PBN. The results showed that this direct activation of M1 neuron terminals in the PBN

substantially enhanced EMG responses in the mylohyoid (Fig. 5e, f) (P < 0.01), indicating that the M1-PBN neural circuit might participate in swallowing function. Consistently, the existence of monosynaptic connections between the M1 and PBN was further confirmed by the presence of GFP-labeled neurons in the PBN, which were detected by anterograde transsynaptic AAV tracing (i.e., AAV2/1-hSyn-Cre was injected into the M1, while AAV2/9-CAG-Dio-EGFP was injected into the PBN) (Fig. 5g)[47], and CTB-555 labeled neurons in the M1, which were detected by injecting Cholera toxin subunit B (CTB)−555 into PBN (Sup Fig. 7). Moreover, this connection was also confirmed by the presence of mCherry-labeled neurons in the PBN after an anterograde vesicular

**Fig. 3 | The activity of M1 L5 excitatory neurons in the contralateral hemisphere was required for the therapeutic effects of EA-CV23 in ischemic mice.**
**a**, **b** Images of c-Fos-positive and CaMKIIα-positive neurons in the contralateral M1 in the PSD and PSD + EA groups. Scale bar, 50 μm (**a**). Excitatory neurons in the contralateral M1 were activated by EA stimulation in PSD model mice. (two-tailed Student's unpaired $t$ test, $N = 6$ per group; $t = 2.592$, $*P < 0.05$) (**b**). **c** Scheme of the experiment for in vivo transcranial two-photon imaging. **d** Representative images of somatic $Ca^{2+}$ transients during EA or sham EA. Scale bar, upper white line: 25 μm; lower black line: 1 min; 200% ($\Delta F/F_0$). **e** Somatic $Ca^{2+}$ transients before (pre), during and after the operation (post-operation) in the EA and sham EA groups (two-tailed Student's unpaired $t$ test, $N = 10$ per group; $t = 0.5679$, $P = 0.5772$; during the operation: $t = 8.457$, $**P < 0.01$; post-operation: $t = 10.68$, $**P < 0.01$).
**f**, **g** Representative traces of water-induced EMG responses in wild-type mice with inhibition of M1 neurons (Control+M1⁻, injection of AAV2/9-CaMKIIα-hM4Di-

mCherry into the M1 and CNO intraperitoneal injection), EA-treated PSD model mice with inhibition of M1 neurons (PSD + EA + M1⁻, PSD induction and EA treatment, injection of AAV2/9-CaMKIIα-hM4Di-mCherry into the M1 and CNO injection), and EA-treated PSD model mice with no inhibition of M1 neurons (PSD + EA + hM4Di, PSD induction and EA treatment, injection of AAV2/9-CaMKIIα-hM4Di-mCherry into the M1; PSD + EA + mCherry+CNO, PSD induction and EA treatment, injection of AAV2/9-CaMKIIα-mCherry as a control virus into the M1 and CNO intraperitoneal injection). Scale bar, 2 s and 0.5 mV (**f**). The ability of EA-CV23 to increase EMG responses in the mylohyoid induced by water delivery was blocked after inhibition of M1 L5 neurons in the different groups (one-way ANOVA, $N = 12$ per group; $F = 10.65$, $**P < 0.01$) (**g**). **h** The effect of EA-CV23 on water consumption was prevented by inhibiting M1 L5 neurons (one-way ANOVA, $N = 11$ per group; $F = 16.63$, $**P < 0.01$). Data are presented as Mean ± SEM. $N$ indicates the number of biologically independent samples, mice per group.

---

stomatitis virus (VSV-mCherry) was injected into the M1 L5 (Fig. 5h). Thus, we proposed that the M1-PBN neural circuit contributed to the EA-CV23-mediated improvement of swallowing function in PSD.

## The M1-PBN-NTS neural circuit is involved in the regulation of swallowing function and mediates the protective effect of EA-CV23 against PSD

The NTS, a subcortical brain region, is well known to regulate the swallowing process[22]. We next explored whether the NTS is connected to the M1-PBN neural circuit and thereby participates in the swallowing process. The role of NTS in swallowing function was first revealed by the fact that inhibition of NTS neurons weakened mylohyoid activity induced by water delivery (Fig. 6a) ($P < 0.01$), and attenuated the EA-CV23-induced increase in expression of c-Fos in the NTS in PSD model mice (Fig. 6b, c) ($P < 0.05$). The existence of monosynaptic connections between the PBN and NTS were validated by the anterograde transsynaptic AAV tracing (i.e., AAV2/1-hSyn-Cre was injected into the PBN, while AAV2/9-CAG-Dio-EGFP was injected into the NTS), and the EGFP-labeled neurons were observed in the NTS (Fig. 6d), suggesting that the NTS might be modulated by the PBN. Then, to explore whether neuronal activity in the NTS is modulated by the M1 and PBN, neuronal spikes were recorded using in vivo multichannel recording following optogenetic activation of M1 neurons and inhibition of PBN neurons. It was found that the frequency of neuronal spikes in the NTS was higher when the M1 was activated, whereas inhibiting PBN neurons prevented the M1 activation-induced increase in neuronal spike frequency in the NTS (Fig. 6e, f) ($P < 0.01$, $P < 0.05$). The virus injection site in the M1 and PBN and the recording site in the NTS were confirmed (Supplementary Fig. 8). These results indicated that both the M1 and PBN contributed to the regulation of neuronal activity in the NTS. Next, to determine whether these brain regions collectively regulate swallowing function, we recorded the M1 activation-induced EMG responses during simultaneous activation or inhibition of both PBN and NTS neurons. It was observed that the EMG responses induced by optogenetic activation of the M1 were decreased after simultaneous inhibition of both PBN and NTS neurons, and the EA-CV23-mediated improvement of EMG responses was also prevented (Fig. 6g) ($P < 0.01$, $P < 0.05$). Nevertheless, simultaneous activation of both PBN and NTS neurons increased the mylohyoid activity induced by optogenetic activation of the M1 in both control mice and PSD mice, while simultaneous activation of these neurons could not increase the EMG responses in PSD mice subjected to EA-CV23 (Fig. 6h) ($P < 0.01$, $P < 0.05$). Thus, these results suggested that the M1, PBN and NTS together regulated swallowing function and that these regulatory effects participated in the therapeutic effect of EA-CV23 on PSD.

Next, we validated whether there were synaptic connections among the M1, PBN, and NTS by using monosynaptic retrograde RV virus lacking a glycoprotein (G protein) (Supplementary Fig. 9)[48], and observed the starter neurons (defined as the PBN neurons projecting to the NTS) in the PBN and few fluorescence-labeled neurons in the M1.

The results suggested that there were few neurons in the PBN that received M1 inputs and projected to the NTS (named the direct M1-PBN-NTS neural circuit) (Fig. 6i). Then, to test the effect of M1-PBN inputs to the NTS, AAV2/1-CaMKIIα-Cre was injected into the M1, and AAV2/9-hSyn-Dio-hM4Di was injected into the PBN to selectively inhibit of PBN neurons innervated by the M1, and a recording electrode was implanted into the NTS. The results showed that the neural activity in the NTS decreased after selective inhibition of PBN neurons innervated by the M1 (Fig. 6j, k). Next, the role of the direct M1-PBN-NTS neural circuit in swallowing function was determined by injecting AAV2/1-CaMKIIα-Cre virus into the M1 and AAV2/9-EF1a-Dio-hChR2-EYFP into the PBN and implanting a fiber into the NTS to selectively activate the terminals of M1-innervated PBN neurons projecting to the NTS[47]. The results showed that EMG responses in the mylohyoid were increased but relatively lower than that induced by M1 activation (Fig. 6l) ($P < 0.01$, $P < 0.05$). Thus, we hypothesized that an indirect M1-PBN-NTS neural circuit in which might one cluster of neurons in the PBN is innervated by the M1 while another projects to the NTS and their signals are relayed within the PBN, might also be involved in swallowing function (Fig. 7). In summary, both direct and indirect M1-PBN-NTS neural circuits could participate in swallowing function and mediate the therapeutic effect of EA-CV23 on PSD.

## Discussion

The mechanism underlying EA-CV23-mediated improvement of swallowing function in PSD has been delineated. Our findings revealed that the excitatory neurons in M1 L5 could control swallowing activity and that the M1-PBN-NTS neural circuit participated in the regulation of the swallowing process and contributed to the effect of EA-CV23 in mice with M1 ischemia-induced PSD. Overall, these results provide valuable insights into how these cortical and subcortical brain regions interact to modulate swallowing activity and why the CV23 acupoint could be selected as a suitable peripheral site for PSD treatment in clinical practice.

The observation that neuronal activity in the M1 was increased by EA-CV23 in PSD model mice was consistent with the increase in activity in PSD patients after transcranial magnetic stimulation (TMS) or transcranial direct current stimulation (tDCS)[49]. Studies on the effect of TMS or tDCS in PSD suggested that activity in the motor cortex was involved in the modulation of swallowing function[32,50,51]. However, these studies did not provide direct evidence for the mechanism by which the M1 regulates the swallowing process. In this study, to investigate the role of M1 neurons in swallowing function, mice with focal ischemia in the M1, which resulted in dysphagia phenotypes, including decreased water consumption and impaired EMG responses in the mylohyoid, were used as PSD model mice[43]. In addition, our study demonstrated the existence of synaptic connections between the M1 and mylohyoid and showed that activation of M1 excitatory neurons could control swallowing activity. The results further suggested that, the PBN and NTS, both of which are the subcortical brain

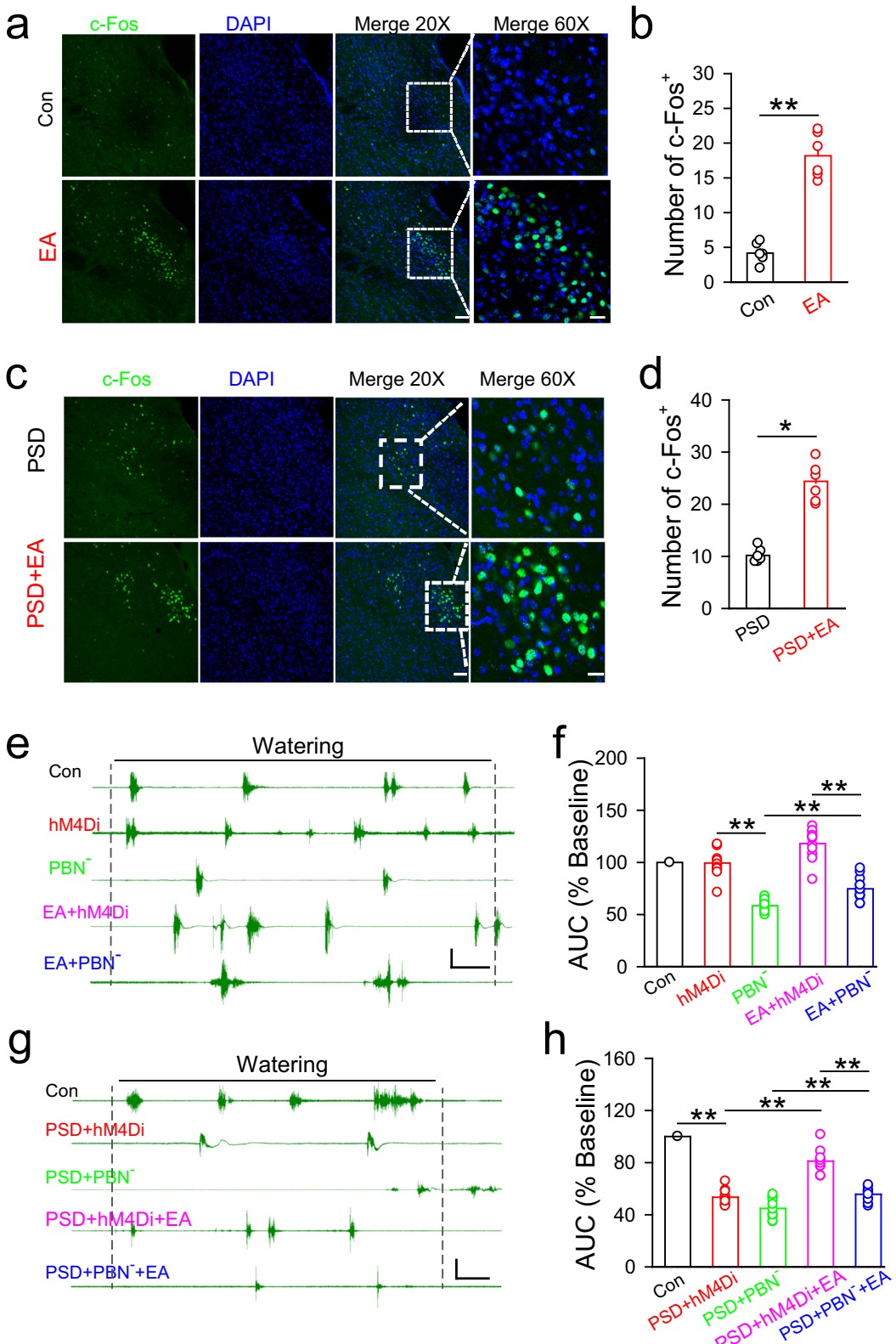

regions, were connected to the M1 and participated together in the regulation of the swallowing process. Notably, although there are various used dysphagia animal models, such as transient middle cerebral artery occlusion[52], Parkinson's disease[53,54], and amyotrophic lateral sclerosis (ALS)[55,56], the pathology of dysphagia in these models is more complicated, and the M1 focal ischemia-induced PSD mouse model is the optimal choice for specifically exploring the role of the M1

in swallowing function. However, it cannot be excluded that the M1, PBN, and NTS brain regions also participated in the regulation of swallowing function in these different dysphagia models. Moreover, while neurons in PBN are recognized to relay the information of taste, feeding, and pain information, little is known about their role in the swallowing process[38]. More studies focusing on the function of the PBN in the swallowing process are needed. This research suggests that

**Fig. 4 | Neurons in the PBN participated in the regulation of swallowing function and the therapeutic effect of EA-CV23 on PSD. a** Representative images of c-Fos expression in the PBN of mice treated with EA or not (Con). Scale bar, 20×: 100 μm; 60X: 50 μm. **b** The expression of c-Fos in the EA group was increased compared to that in the Con group (two-tailed Student's unpaired t test, N = 6 per group; t = 9.648, **P < 0.01). **c, d** The number of c-Fos positive neurons in the PBN of PSD model mice treated with EA (PSD + EA) or not (PSD). Green: c-Fos positive neurons. Blue: DAPI (Nuclei). Scale bar, 20×: 100 μm, 60×: 50 μm (**c**). The expression of c-Fos in the PBN was enhanced in the PSD + EA group compared to the PSD group (two-tailed Student's unpaired t test, N = 6 per group; t = 8.601, **P < 0.01) (**d**). **e, f** Representative traces of EMG responses induced by water delivery in mice from the following groups: control (Con), hM4Di injection (hM4Di, injection of AAV2/9-CaMKIIα-hM4Di-mCherry into the PBN), and hM4Di and CNO injection (PBN⁻, injection of AAV2/9-CaMKIIα-hM4Di-mCherry into the PBN and CNO intraperitoneal injection), hM4Di injection and EA treatment (EA + hM4Di, EA treatment, and injection of AAV2/9-CaMKIIα-hM4Di-mCherry into the PBN), and hM4Di and CNO injection and EA treatment (EA + PBN⁻, EA treatment and injection of AAV2/9-

CaMKIIα-hM4Di-mCherry into the PBN and CNO intraperitoneal injection) groups. Scale bar, 1 s and 0.5 mV (**e**). The AUC of EMG responses was decreased after inhibition of PBN neurons (hM4Di vs. PBN⁻), and the effect of EA was also attenuated (EA + hM4Di vs. EA + PBN⁻) (one-way ANOVA, N = 12 per group; F = 62.83; post hoc Tukey's test: **P < 0.01) (**f**). **g, h** Representative traces of water-induced EMG responses in mice from the following groups: control (Con), PSD (PSD + hM4Di, PSD induction and injection of AAV2/9-CaMKIIα-hM4Di-mCherry virus into the PBN), PSD and inhibition of PBN neurons (PSD + PBN⁻, PSD induction and injection of AAV2/9-CaMKIIα-hM4Di-mCherry virus into the PBN and CNO intraperitoneal injection), PSD and EA treatment (PSD + hM4Di + EA, PSD induction, EA treatment, and injection of AAV2/9-CaMKIIα-hM4Di-mCherry virus into the PBN), and PSD with EA treatment and inhibition of PBN neurons (PSD + PBN⁻ + EA, PSD induction, EA treatment, injection of AAV2/9-CaMKIIα-hM4Di-mCherry virus into the PBN and CNO intraperitoneal injection) groups. Scale bar, 1 s and 0.5 mV (**g**). The effect of EA treatment in PSD model mice was blocked by inhibiting PBN neurons (one-way ANOVA, N = 11 per group; F = 145.5, **P < 0.01) (**h**). Data are presented as Mean ± SEM. N indicates the number of biologically independent samples, mice per group.

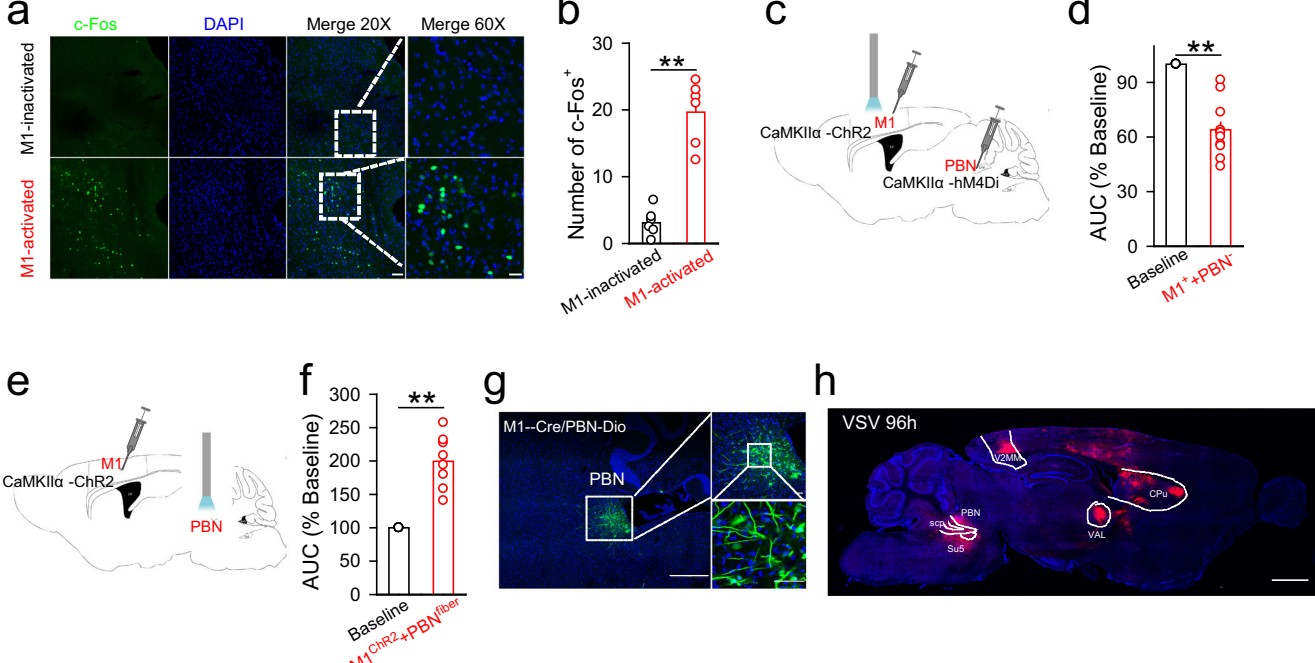

**Fig. 5 | The M1-PBN neural circuit was involved in the regulation of swallowing function. a, b** The expression of c-Fos-positive neurons in the PBN following optogenetic activation of M1 neurons (M1-activated, injection of AAV2/9-CaMKIIα-ChR2-mCherry into the M1 and blue light stimulation) or not (M1-inactivated, injection of AAV2/9-CaMKIIα-mCherry into the M1). Green: c-Fos-positive neurons. Blue: Nuclei. Scale bar, 20×: 100 μm, 60×: 50 μm (**a**). M1 neurons activation increased the expression of c-Fos in the PBN (two-tailed Student's unpaired t-test, N = 6 per group; t = 7.717, **P < 0.01) (**b**). **c, d** Diagram showing optogenetic activation of M1 neurons and inhibition of PBN neurons. AAV2/9-CaMKIIα-ChR2-mCherry was injected into the M1, and light stimulation was applied, while AAV2/9-CaMKIIα-hM4Di-mCherry was injected into the PBN and CNO was administered by intraperitoneal injection (**c**). Inhibition of excitatory neurons in the PBN attenuated the M1 activation-induced EMG responses in the mylohyoid (M1⁺ + PBN⁻) (two-tailed

Student's unpaired t test, N = 12; t = 8.828, **P < 0.01) (**d**). **e, f** Schematic diagram showing selective activation of the M1-PBN neural circuit. AAV2/9-CaMKIIα-ChR2-mCherry was injected into the M1, and blue light was delivered to the PBN via an implanted fiber (**e**). Selectively activating the terminals of M1 neurons projecting to the PBN increased the EMG response (M1^ChR2+PBN^fiber). (two-tailed Student's unpaired t test, N = 16; t = 7.717, **P < 0.01) (**f**). **g** Left: EGFP-positive neurons were detected in the PBN by monosynaptic tracing with AAV2/1-hSyn-Cre was injected into the M1, while AAV2/9-CAG-Dio-EGFP was injected into the PBN. Scale bar, 500 μm. Right: the enlarged versions of the images on the left. N = 5. Scale bar, 100 μm. **h** The distribution of trans-synaptically labeled neurons at 96 h. Anterograde tracing was performed by injecting with VSV-mCherry into the M1. N = 5. Scale bar, 100 μm. Data are presented as Mean ± SEM. N indicates the number of biologically independent samples, mice per group.

the excitatory neurons contribute to this circuit, while the role of inhibitory neurons, such as agouti gene-related protein (AgRP) neurons in the PBN[57], and gamma aminobutyric acid (GABA)ergic neurons in the NTS[58], needs further investigation. Furthermore, studies are required to investigate how neurons within the PBN transfer signals from the M1 to NTS.

In this study, the M1-PBN-NTS neural circuit was proposed to be an important pathway in dysphagia. We first optogenetically activated M1 neurons and found the increased c-Fos expression in the PBN.

Moreover, inhibition of PBN neurons could attenuate M1 activation-induced EMG responses in the mylohyoid. Furthermore, selective optogenetic activation of the terminals of M1 neurons projecting to PBN enhanced EMG responses. These results suggested that the M1 could regulate the PBN and may have concurrently modulated swallowing function. Accordingly, a combination of CTB, AAV2/1-Cre+Dio, and VSV tracing was used to validate the existence of M1-PBN projections. Next, to demonstrate the existence of the M1-PBN-NTS neural circuit, RV-ΔG system was used to confirm the presence of synaptic

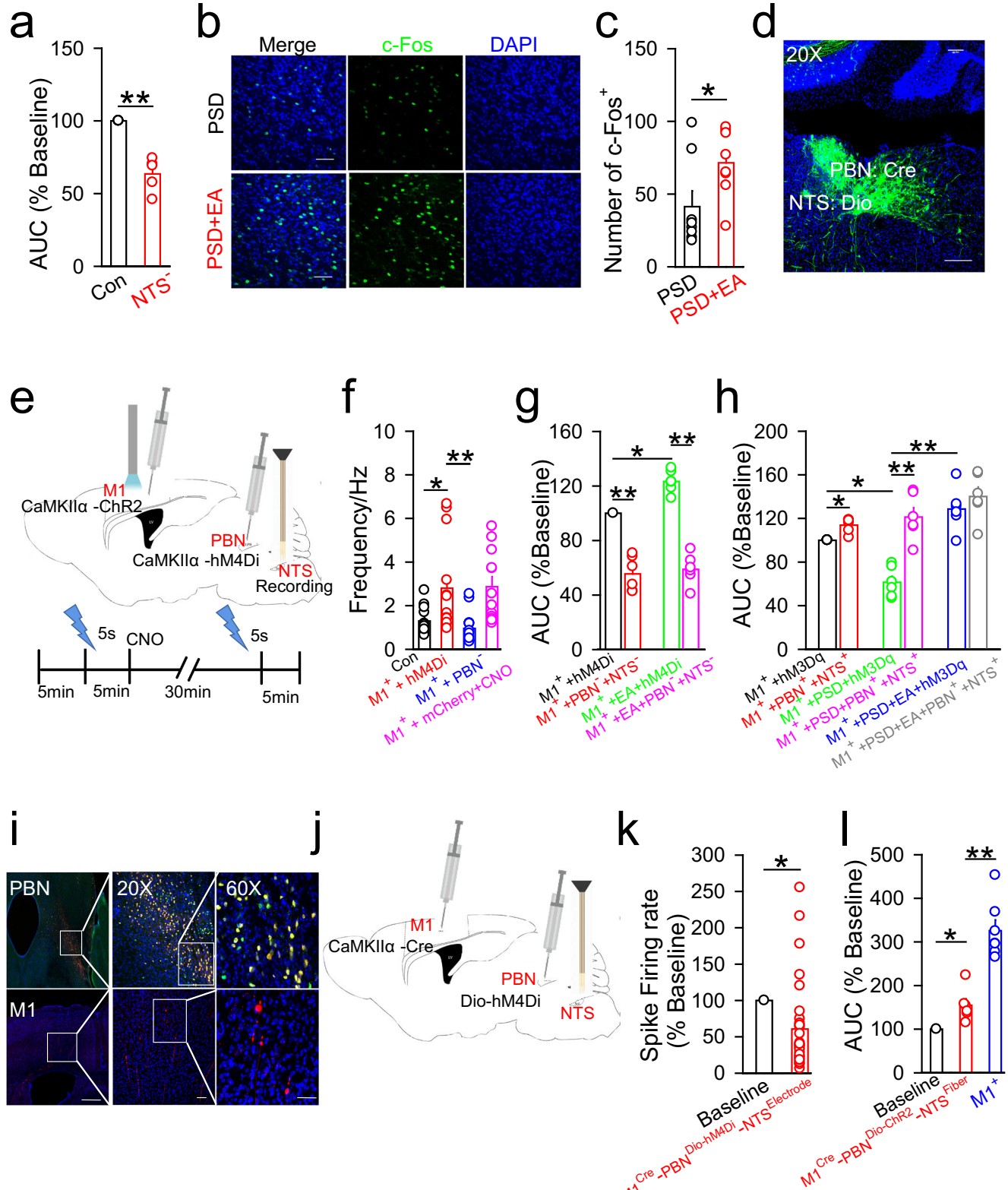

connections among these brain regions. In the experiment using AAV2/1+Dio, the fluorescent neurons in the PBN (Fig. 5g), and NTS (Fig. 6d) were observed; however, the possibility of a PBN-M1 or NTS-PBN neural circuit could not be excluded considering the potential retrograde property of these viruses. Although the existence of a reverse NTS-PBN-M1 neural circuit was not ruled out by the above experiments, the function of the M1-PBN-NTS neural circuit was investigated in different experiments. In vivo electrophysiological recording showed that inhibiting PBN neurons prevented the M1 activation-induced increase in neuronal spike frequency in the NTS (Fig. 6e, f), and that inhibition of PBN neurons innervated by the M1 decreased neural activity in the NTS (Fig. 6k). Furthermore, selective optogenetic activation of the terminals of M1-innervated PBN neurons projecting to the NTS could induce EMG responses in the mylohyoid (Fig. 6l). Nevertheless, there is likely an NTS-PBN-M1 neural circuit that participates in transmitting information from the NTS to M1. More

**Fig. 6 | The M1-PBN-NTS neural circuit was involved in the regulation of swallowing function and mediated the protective effect of EA-CV23 against PSD.**
**a** EMG responses induced by water delivery were impaired in the NTS neurons inhibition group (NTS⁻, injection of AAV2/9-CaMKIIα-hM4Di-mCherry in the NTS and CNO intraperitoneal injection) compared to the Con group (Con: injection of AAV2/9-CaMKIIα-hM4Di-mCherry into the NTS) (two-tailed Student's unpaired $t$ test, $N = 5$ per group; $t = 7.022$, **$P < 0.01$). **b, c** Representative images of c-Fos expression in the NTS of PSD model mice (PSD), and PSD model mice treated with EA (PSD + EA). Scale bar, 100 μm (**b**). The expression of c-Fos in the NTS was increased by EA-CV23 in PSD model mice (two-tailed Student's unpaired $t$ test, $N = 8$ per group; $t = 2.210$, *$P < 0.05$) (**c**). **d** The EGFP-labeled neurons in the PBN were observed by monosynaptic tracing with AAV2/1-hSyn-Cre was injected into the PBN, and AAV2/9-CAG-Dio-EGFP was injected into the NTS. Scale bar, 100 μm.
**e** Schematic diagram showing the strategy used to determine whether neuronal activity in the NTS was modulated by the M1 and PBN. AAV2/9-CaMKIIα-ChR2-mCherry virus was injected into the M1, AAV2/9-CaMKIIα-hM4Di-EGFP was injected into the PBN and a multichannel electrode was implanted into the NTS. The timeline shown below demonstrates the experimental procedure. First, 5 s of blue light was delivered to optogenetically activate the M1 neurons, and then neuronal activity recorded in the NTS was recorded for 5 min. 30 min after CNO injection to inhibit PBN neurons, another light stimulus was given, and neuronal activity was recorded simultaneously for another 5 min. **f** Neuronal activity in the NTS was significantly increased after activation of M1 neurons (M1⁺ + hM4Di, injection of AAV2/9-CaMKIIα-ChR2-mCherry into the M1, blue light stimulation, and injection of AAV2/9-CaMKIIα-hM4Di-mCherry into the PBN), compared to that at baseline (Con, injection of AAV2/9-CaMKIIα-ChR2-mCherry into the M1 and injection of AAV2/9-CaMKIIα-hM4Di-mCherry into the PBN), and this increase was blocked by inhibition of PBN neurons (M1⁺ + PBN⁻, injection of AAV2/9-CaMKIIα-ChR2-mCherry into the M1, blue light stimulation, injection of AAV2/9-CaMKIIα-hM4Di-mCherry into the PBN and CNO intraperitoneal injection). To control for the effect of the hM4Di-expressing virus, an M1⁺ + mCherry+CNO group (injection of AAV2/9-CaMKIIα-ChR2-mCherry into the M1, blue light stimulation, injection of AAV2/9-CaMKIIα-mCherry as a control virus into the PBN and CNO intraperitoneal injection) was included. A total of 38 units were recorded; 10 units were observed in the normal, M⁺ + hM4Di, and M1⁺ + PBN⁻ groups respectively, while 8 units were observed the in M1⁺ + mCherry+CNO group (one-way ANOVA, $N = 8$ per group; $F = 6.831$, $P = 0.0006$, Con vs. M1⁺ + hM4Di, *$P < 0.05$; M1⁺ + hM4Di vs. M1⁺ + PBN⁻). **g** Inhibiting neuronal activity in both the PBN and NTS decreased the EMG responses induced by optogenetic activation of M1 neurons, M1⁺ + hM4Di (injection of AAV2/9-CaMKIIα-ChR2-mCherry into the M1, blue light stimulation, injection of AAV2/9-CaMKIIα-hM4Di-mCherry into the PBN and NTS) vs. M1⁺ + PBN⁻ + NTS⁻ (injection of AAV2/9-CaMKIIα-ChR2-mCherry into the M1, blue light stimulation, injection of AAV2/9-CaMKIIα-hM4Di-mCherry into the PBN and NTS and CNO intraperitoneal injection), and prevented the EA-CV23-induced increase in EMG responses, M1⁺ + hM4Di vs M1⁺ + EA + hM4Di (EA treatment, injection of AAV2/9-CaMKIIα-ChR2-mCherry into the M1, blue light stimulation, and injection of AAV2/9-CaMKIIα-hM4Di-mCherry into the PBN and NTS), and M1⁺ + EA + hM4Di vs.

M1⁺ + EA + PBN⁻ + NTS⁻ (EA treatment, injection of AAV2/9-CaMKIIα-ChR2-mCherry into the M1, blue light stimulation, and injection of AAV2/9-CaMKIIα-hM4Di-mCherry into the PBN and NTS and CNO intraperitoneal injection) (one-way ANOVA, $N = 6$ per group; $F = 74.20$, **$P < 0.01$). **h** Activation of both PBN and NTS increased EMG responses in the non-PSD and PSD groups. M1⁺ + hM3Dq (injection of AAV2/9-CaMKIIα-ChR2-mCherry into the M1, blue light stimulation, and injection of AAV2/9-CaMKIIα-hM3Dq-mCherry into the PBN and NTS) vs. M1⁺ + PBN⁺ + NTS⁺ (injection of AAV2/9-CaMKIIα-ChR2-mCherry into the M1, blue light stimulation, and injection of AAV2/9-CaMKIIα-hM3Dq-mCherry into the PBN and NTS and CNO intraperitoneal injection), M1⁺ + PSD + hM3Dq (PSD induction, injection of AAV2/9-CaMKIIα-ChR2-mCherry into the M1, blue light stimulation, and injection of AAV2/9-CaMKIIα-hM3Dq-mCherry into the PBN and NTS) vs. M1⁺ + PSD + PBN⁺ + NTS⁺ (PSD induction, injection of AAV2/9-CaMKIIα-ChR2-mCherry into the M1, blue light stimulation, and injection of AAV2/9-CaMKIIα-hM3Dq-mCherry into the PBN and NTS and CNO intraperitoneal injection); and prevented the EA treatment-induced increase in EMG responses, M1⁺ + PSD + hM3Dq vs. M1⁺ + PSD + EA + hM3Dq (PSD induction, EA treatment, injection of AAV2/9-CaMKIIα-ChR2-mCherry into the M1, blue light stimulation, and injection of AAV2/9-CaMKIIα-hM3Dq-mCherry into the PBN and NTS), and M1⁺ + PSD + EA + hM3Dq vs. M1⁺ + PSD + EA + PBN⁺ + NTS⁺ (PSD induction, EA treatment, injection of AAV2/9-CaMKIIα-ChR2-mCherry into the M1, blue light stimulation, injection of AAV2/9-CaMKIIα-hM3Dq-mCherry into the PBN and NTS and CNO intraperitoneal injection) (one-way ANOVA, $N = 6$ per group; $F = 18.30$, **$P < 0.01$). **i** The distribution of labeled neurons in the PBN and M1 assessed by using monosynaptic retrograde RV-ΔG. Top: the white box indicates the starter neurons (yellow) in the PBN projecting to the NTS, the middle is an enlarged version of the figure on the left, and the starter neurons on the right indicates neurons expressing both the helper (EGFP) and RV (Red). Bottom: the distribution of RV-infected retrograde neurons in the M1 projecting from the PBN. The image on the right is an enlarged version of the image in the white box in the middle. Scale bar, left: 1000 μm; middle: 100 μm; right: 50 μm. **j, k** Diagram showing the strategy used to record neuronal spikes in the NTS following inhibition of M1 projection neurons in the PBN. AAV2/1-CaMKIIα-Cre was injected into the M1, and AAV2/9-hSyn-Dio-hM4Di-mCherry was injected into the PBN, CNO was delivered by intraperitoneal injection, and a recording electrode was implanted into the NTS (M1^Cre-PBN^Dio-hM4Di-NTS^Electrode) (**j**). Inhibition of M1 projection neurons in the PBN could decrease the neuronal activity in the NTS. A total of 31 units were recorded and analyzed (two-tailed Student's paired $t$ test, N = 5; $t = 3.614$, **$P < 0.01$) (**k**). **l** EMG responses were induced by activation of direct M1-PBN-NTS (M1^Cre-PBN^Dio-ChR2-NTS^Fiber, injection of AAV2/1-CaMKIIα-Cre into the M1 and AAV2/9-EF1a-Dio-hChR2-EYFP into the PBN, and fiber implantation into the NTS and blue light stimulation), and this response was smaller than that observed following activation of M1 neurons (M1⁺, injection of AAV2/9-CaMKIIα-ChR2-mCherry into the M1 and blue light stimulation) (one-way ANOVA, $N = 6$ per group; $F = 50.09$; baseline vs. M1^Cre-PBN^Dio-ChR2-NTS^fiber, *$p < 0.05$; M1^Cre-PBN^Dio-ChR2-NTS^fiber vs. M1⁺, **$P < 0.01$). Data are presented as Mean ± SEM. $N$ indicates the number of biologically independent samples, mice per group.

---

studies focused on improving our understanding of how the M1, PBN, and NTS interact with each other and modulate the swallowing process after EA-CV23 are needed.

The VFSS, FEES, SSA, and WST are mostly applied in clinical practice to evaluate swallowing function in humans[59–61], but these techniques have not been widely used in rodent experiments. Recently, Lever's laboratory at the University of Missouri School of Medicine has developed a device performing transoral endoscopy (for FEES) and videofluoscopic swallowing study (for VFSS) in rodents[62,63]. In this study, EMG recording, through which most of our results were obtained, was used to test neuromuscular function by measuring the muscle response to a nerve stimulation[64]. The AUC (calculated for most results) reflected the number and diameter of the activated muscle fibers; thus, the EMG responses could reflect the activity of the recorded muscles[64]. This method is comparable with the PsEMG methods used in clinical practice but targets on the presumed mylohyoid in this study. The water consumption test only matches the ability of the WST to evaluate drinking water ability to some extent. Therefore, although the VFSS, FEES, SSA, WST, and PsEMG

assessments have proven the effect of stimulation at CV23 on swallowing function in clinical practice, a limitation of this study is that it does not provide more direct evidence. We are trying to use the FEES and VFSS methods, reported by Teresa E. Lever's laboratory[62,65]. Our preliminary results from transoral endoscopy results showed the asymmetric and incomplete closure of the vocal fold in PSD model mice, which was attenuated by EA-CV23.

Swallowing is a complex sensorimotor process involving the coordinated contraction of the tongue, larynx, pharynx, and esophagus muscles[66], and can be subdivided into two phases, the oropharyngeal and esophageal phases[22,67,68]. With the aim of exploring the mechanism underlying the ability of EA-CV23, located in the mylohyoid[40], to improve swallowing function in PSD, mylohyoid activity was recorded. The mylohyoid (superior and deep to the anterior belly of the digastric muscle), in addition to the geniohyoid, and anterior digastric muscles, is active during the initiation of the swallowing process by pulling the larynx upward[68]. The nearby muscles are superimposed with others, including the geniohyoid, mylohyoid, and hyoglossus. It is likely that the responses of muscles near

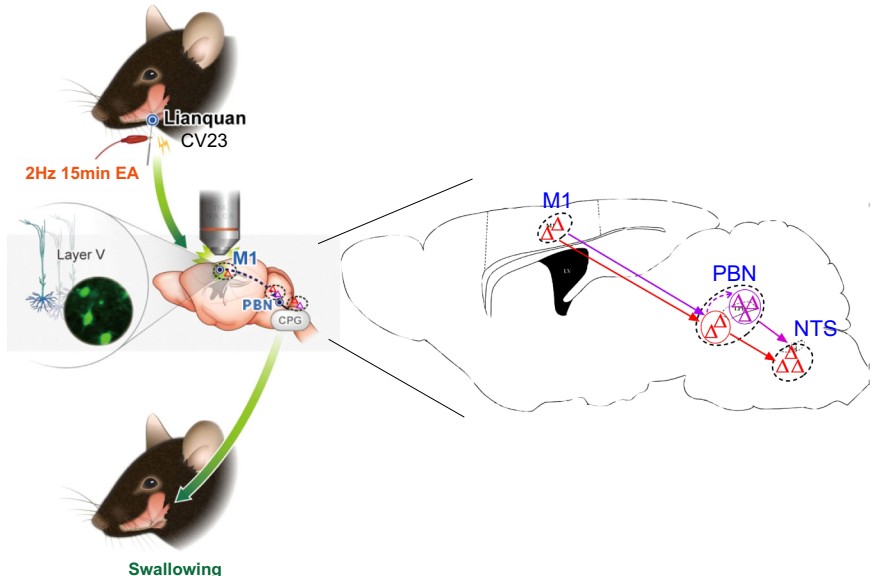

**Fig. 7 | Diagram of the mechanism underlying the effect of EA-CV23 on PSD treatment.** Excitatory neurons in the contralateral M1 L5 are required for EA-CV23-mediated alleviation of swallowing dysfunction in PSD model mice. This modulatory involves two forms of the M1-PBN-NTS neural circuit, i.e., a direct M1-PBN-NTS circuit (red line) and a relay circuit between two clusters of neurons within PBN (purple line).

the mylohyoid could also be collected during EMG recording, and affected by EA-CV23 stimulation. However, the role of other nearby muscles in M1-PBN-NTS neural circuit-mediated swallowing activity was not tested in this study. The hypoglossal nerve, which innervates to the geniohyoid and hyoglossus, was demonstrated to be activated by EA-CV23[22,43]. The activation of sensory nerves, including the hypoglossal nerve and mandibular division of the trigeminal nerve (innervates to the mylohyoid), plays important roles in triggering the swallowing process by transmitting sensory swallowing information to the brainstem and then activating the motor cortex[69–71], and this mechanism may underlie EA-CV23-induced M1 activation. It was previously found that EA-CV23 causes the release of peripheral neurotransmitters, such as substance P (SP), and thus regulates the excitability of primary afferents of local sensory nerves[43]. As another form of peripheral pharyngeal stimulation, PES, which is achieved by a nasogastric tube containing a pair of catheter-mounted electrodes placed on the base of the tongue and posterior pharyngeal wall through the nasal cavity[72], is widely used for the treatment of PSD in clinical practice. This method stimulates a region close to the CV23 acupoint and can increase the activity of M1 as well[6,32]. The M1-PBN-NTS neural circuit, which was found to underlie the effect of EA-CV23, is probably also involved in effect of PES, but this has not yet been confirmed. Moreover, EA, which is safe, easy to operation, convenient and inexpensive, might be a more welcomed therapy for dysphagia than PES.

In conclusion, our study demonstrated that electrical stimulation at the CV23 acupoint, as a peripheral stimulation strategy, could improve the swallowing function in PSD model mice through the activation of motor cortex inputs to the NTS through the PBN. Our study provides an experimental basis for the application of EA-CV23 for the treatment of PSD by addressing how motor cortex activation benefits recovery of swallowing function via a subcortical pathway, suggesting that acupuncture treatment could be used as an effective therapeutic intervention to improve swallowing function. These findings not only elucidate a critical pathway for understanding how the motor cortex controls the swallowing process, but also suggest a potential valuable treatment strategy for swallowing-related disorders.

## Methods

### Animals

Male C57BL/6 mice or GAD67-GFP transgenic mice (2–3 months old, 20–25 g) were used. All mice were housed in home cages at a temperature of 23 ± 2 °C and humidity of 50 ± 5% and had access to food and water ad libitum (unless mentioned). The animals were housed on a controlled 12/12 h light/dark cycle with lights on from 7:00 to 19:00. The animals were randomly allocated to an experimental or a control group, and the experimenters were blinded to the groups. All surgery and experimental procedures in this study were performed in accordance with the guidelines of the Committee for Care and Use of Research Animals of Guangzhou University of Chinese Medicine (No. 20170303).

### Retrograde tracing from the mylohyoid muscle

For virus-mediated retrograde trans-multisynaptic tracing, the PRV-CMV-EGFP tracer packaged by BrainVTA was used, as previously described[73]. The titer of PRV was -2 × 10^9 pfu/ml. Adult male C57BL/6 mice were anesthetized by intraperitoneal injection of pentobarbital sodium (Merck, Germany. 80 mg/kg), and then injected with 1 μl of PRV-CMV-EGFP into two sites of the mylohyoid, located between the mandible and hyoid bone, for a total of 2 μl per mouse. After 48, 72, and 96 h, the mice were sacrificed group by group via anesthetization with pentobarbital sodium and transcardial perfusion with 0.9% saline followed by 4% paraformaldehyde (PFA). The brains were removed, postfixed in 4% PFA overnight, and then cryoprotected in 30% sucrose solution for 48 h at 4 °C. Brain tissues were sliced into 40 μm sections using a freezing microtome (HM525, Thermo, USA) and imaged with a confocal microscope (A1, Nikon, Japan).

### In vivo EMG recording

The EMG recording is used to assess the neuromuscular function by measuring the muscle response to nerve stimulation and is comparable to the PsEMG recording in clinical practice but more likely suitable for targeting the internal muscle presumed mylohyoid[74]. The mylohyoid is involved in the initiation of swallowing and functions as a laryngeal elevator, pulling the larynx upward. Thus, EMG responses to some extent reflect swallowing function[66,75,76]. To record EMG

responses in the awake mice, the mice were first anesthetized with isoflurane, taken out from the anesthesia box and quickly fixed to a mouse adaptor in the supine position. Then, a pair of custom stainless-steel wire electrodes (~15 mm in diameter) were inserted unilaterally into the left mylohyoid through the skin to a depth of ~0.5 cm, and a tube (0.85 × 0.42 mm PE tube) for water delivery was positioned against the mouse's hard palate. To prevent the mice from struggling throughout the whole recording process, an anesthetic mask connected to isoflurane was available to the mouse's nostrils. The M1 activation-induced EMG responses were recorded by optogenetically stimulating M1 neurons transfected with AAV2/9-CaMKIIα-ChR2-mCherry (BrainVTA, titer of $10^{12}$). Optical fibers (Fiblaser, China) in a ceramic ferrule were implanted unilaterally with in the M1 (AP: −0.12 mm, ML: 1.03 mm and DV: −0.85 mm). The mice were allowed to recover for at least 7 d after fiber implantation. Then, the mice were exposed to 473 nm light controlled by an optogenetic controller (PlexBright, Plexon, USA) and Radiant Software (Plexon, USA). The delivered light was synchronized with the EMG recordings. In the experiments in which water-induced EMG responses were recorded[77], a total of 20 μl water was delivered by a microinjection pump (HARVARD, USA) at a speed of 2 μl/s. To prevent the water from running out from the nose, the mice were placed in a semireclined position, and water was given in at least 1 min intervals to ensure the mice had enough time to swallow the water. All EMG signals were acquired with Spike2 software (CED, UK). The data were digitized at 5 kHz with a Power 1401 digitizer (CED, UK) and bandpass filtered at 0.1-1 kHz with a 1902 differential AC amplifier (CED, UK). The AUC (area under the curve) was calculated for a 10 s after the onset of water delivery and for 5 s window after the onset of optical stimulation for analysis.

## Balloon pressure detection

To verify the reliability of the EMG recordings for evaluating swallowing function, as shown in Fig. 1d–g, a homemade balloon was used to assess M1 activation-induced swallowing responses in awake mice during EMG recording[78]. First, the mice were anesthetized with isoflurane and quickly fixed in the supine position. Then, a balloon of an appropriate size for the pharyngeal airway was placed against the root of the tongue and near the hard palate. The balloon, which had a diameter of ~3 mm, allowed space for breathing. The balloon was connected to a pharyngeal pressure transducer, called a baroreceptor, which was linked to a three-way valve, and a sphygmomanometer was first used to adjust the pressure of the baroreceptor. Then, the sphygmomanometer was replaced with a syringe filled with water. Meanwhile, the EMG recording electrodes were inserted into the mylohyoid. A machine (LabChart, MATLAB, America) connected to the baroreceptor was used to measure the pressure of the balloon when the mice were exposed to blue light stimulation; thus, the pressure of the balloon and EMG responses were measured simultaneously during blue light stimulation in awake mice without apparent struggle.

## Establishment of the PSD mouse model

The mice were placed in an anesthesia induction box containing 4% isoflurane in an oxygen/air mixture until anesthetized, and then transferred to an anesthetic mask to maintain anesthesia with 2% isoflurane provided by an animal anesthesia machine (R500, RWD, China). Then the tail was soaked in warm water at 38 ± 2 °C to expand the tail vein, and the tail vein was injected with 15 mg/ml RB solution (diluted with 0.9% saline, Sigma, USA) as previously described[45,79]. The coordinates of the M1 (AP: −0.12 mm, ML: −1.03 mm) and PFC (AP: −0.3 mm, ML: −1.75 mm) were selected based on the PRV labeling regions and the *Paxinos and Franklin Mouse Brain Atlas* (second edition). A laser beam with a 1 mm diameter and 532 nm wavelength (Laserware, China) was stereotactically positioned in the middle of the thin targeted region. All other brain regions except for the targeted area were covered with shading paper to avoid unintended peripheral

illumination. After ~8 min of illumination, the scalp was sutured, and the mice were allowed to fully recover from anesthesia before being returned to their home cage.

## Laser speckle contrast imaging

After the photothrombosis was complete, the cerebral blood perfusion was examined by a Laser Speckle Blood Monitor (PSI-ZR, Pericam PSI, Sweden)[80]. Briefly, the mice were anesthetized with 2% isoflurane in an oxygen/air mixture. A charge-coupled device camera was placed ~11 cm above the head. The intact skull surface was illuminated by a laser diode to allow laser penetration through the brain. Laser speckle imaging was performed with a sampling frequency of 5 Hz, resolution of 1386 × 1034 pixels, and a zoom size of 1.5 × 1.5 cm. The scanning lasted for 3 min, and the average value in blood perfusion units was calculated.

## Electroacupuncture procedure

Mice were subjected to EA-CV23 24 h after M1 ischemia[43,44]. At this time point, EA significantly improves swallowing function and blood flow[43,44]. Therefore, 24 h post stroke induction, mice were anesthetized with 2% isoflurane in an oxygen/air mixture. EA was performed by obliquely inserting (toward the tongue root) a 0.16 × 10 mm unipolar stainless-steel needle (Suzhou Medical Appliance Factory, China) at a depth of ~0.5 cm into the CV23 acupoint, which is located at the presumed mylohyoid. It should be noted that there are many superimposed muscles in mice, including the digastric, mylohyoid, geniohyoid, and hyoglossus, in mice. However, there is no specific marker distinguishing these different muscles; thus, the location of the needle was confirmed by dye. The dye was injected at the same location as the acupuncture needle insertion site, as shown in Sup Fig. 1a. Targeting of the presumed mylohyoid was confirmed by dissecting these muscles as previously described[12,81]. Intermittent pulses with a frequency of 2 Hz and intensity of 1 mA were delivered for 15 min by a Master 8 stimulator instrument (AMPI, Israel)[44]. These parameters (2 Hz/15 min) were selected based on the evidence from our clinical study and previous reports[44,82]. Regarding intensity, a recent paper published in *Neuron* showed that 1.0 mA intensity stimulation at another widely used acupoint, ST36, produced an anti-inflammatory effect[83]. In addition, in mice, 1.0 mA stimulation can lead to the visible contraction of muscles near the acupoint during stimulation, which is important for estimating the effectiveness of this treatment. Sham EA was performed following the same procedure but with a nonelectrical wooden toothpick instead of a stainless-steel needle. In both clinical trials and basic animal studies, this method has been adopted as a sham intervention[84–87].

## Water consumption test

After establishing the PSD mouse model, we evaluated swallowing activity by the water consumption test[88]. This test can be used to mimic Kubota water swallowing Test, which is used in clinical practice, to evaluate water drinking ability. The average daily amount of food and water consumed by a 25 g adult mouse are 3–5 g and 4 mL, respectively[79]. Sipper tube bottles were fabricated by modifying from a 50 mL centrifuge tube with a silicone stopper. Then, a metal spout with a stainless steel ball and a stainless steel pressure bar was connected to a leakproof sealing ring to prevent the water from leaking. Each mouse was fed individually in a separate cage during the test. The bottle in each cage was removed 24 h prior to the water consumption test. The water in the bottles (pH, ~7.4; hardness,108 mg/l; chromaticity, <5) was filtered from the tap water. The volume of water consumed by each mouse within 4 min after the moment when the tongue touched the water was measured, and there was nearly no spontaneous water leakage from the bottle. The consumption of water was measured by subtracting the volume of water remaining after 4 min from the initial volume of water.

## Immunofluorescence

The mice were anesthetized by intraperitoneal injection of 1.25% avertin to assess c-Fos expression after EA stimulation for 1 h. The mice were perfused with 0.9% saline and 4% PFA. The mouse brain was fixed in 4% PFA overnight and dehydrated with 30% sucrose solution for 3 d. The brain tissues were embedded in Optimal Cutting Temperature (OCT) compound (Tissue-Tek) and cut into 40 μm with a freezing microtome (Thermo, USA). The brain was sectioned to. Coronal brain slices were rinsed three times with 0.01 M PBS for five minutes, blocked with blocking solution containing 1% BSA (Macklin, B885114, China) and 0.3% Triton X-100 (Biosharp, BL935A, China) blocking solution for 1.5 h at 37 °C and incubated with a rabbit polyclonal anti-c-Fos antibody (1:500 dilution, CellSignaling Technology, 2250 s, USA) at 4 °C overnight, followed by Alexa Fluor 488-conjugated goat anti-rabbit secondary antibody (1:500 dilution, ThermoFisher, A11034, USA) or Alexa Fluor 594-conjugated donkey anti-rabbit secondary antibody (1:500 dilution, ThermoFisher, A21207, USA) for 1 h at 37 °C. The brain sections were stained with nuclear staining solution for 5-10 min (DAPI, 1 μg/5 ml, Sigma, D9542 USA) and rinsed three times with 0.01 M PBS as mentioned above. After that, the slices were placed on the slides and sealed with mounting medium (50% glycerol anhydrous in PBS, Biofroxx, 1280ML100, Germany). To determine the type of c-Fos- positive neurons (as shown in Fig. 4a, b and Supplementary Fig. 3c–d), mice transfected with AAV2/9-CaMKIIα-mCherry or GAD67-GFP transgenic mice, in which GABAergic neurons were labeled with GFP under the control of the GAD67 promoter, were used. The sections were imaged with a confocal microscope (Nikon, Japan) at 20× or 40× magnification. High-resolution images were obtained at 60×. Confocal images were acquired, and cell counts were performed using ImageJ software (1.52a, National Institutes of Health, USA). The expression of c-Fos was manually quantified in various brain regions by an observer blinded to the experimental conditions.

## In vivo transcranial two-photon imaging

The activity of individual neurons in the M1 during EA stimulation was assessed using an in vivo two-photon imaging technique[89]. The mice were injected with AAV2/9-CaMKIIα-GCaMP6s into the M1 (AP: −0.12 mm, ML: −1.03 mm, DV: −1.10 mm), and the skin was sutured, the mice were placed in their home cages for at least 21 d prior to imaging to allow virus expression. Before imaging, the mice were anesthetized with 1.25% avertin (0.2 mL per 10 g, Sigma, St. Louis, MO, USA). The hair on the skull was shaved with a razor, and both eyes were lubricated with erythromycin eye ointment to protect against drying. The head was fixed in a homemade stereoscopic apparatus comprising two sticks that formed a groove for the water lens. An open-skull observation window (2 mm in diameter) was made in the skull with a drill. The hole was sealed with a cover slip and glue, and the animal was then fixed on a customized stage under a two-photon microscope (Nikon, Japan). A water-immersion objective (25X, Nikon, Japan) was used for observation. GFP was excited by a 920 nm laser light. Three sessions, i.e., the baseline signal (3 min before the stimulation of EA/sham EA, called Pre), stimulation signal (during 3 min of EA/sham EA, called During), and post-stimulation (3 min after the stimulation of EA/sham EA, called post-operation), were recorded. The images were analyzed by image software (NIS-Elements AR Analysis). The analysis of signals was conducted by an independent observer in a blinded manner.

## Fiber photometry recording

To investigate population neuronal activity in different brain regions (M1 and PFC) in response to EA stimulation, we used fiber photometry recording to measure Ca²⁺ signals. The mice were unilaterally injected with 200 nl AAV2/9-CaMKIIα-GCaMP6s into the M1 (AP: −0.12 mm, ML: 1.03 mm and DV: −1.10 mm) or PFC (AP: 0.3 mm,

ML: 1.75 mm and DV: −2.45 mm), and an optical fiber was implanted 100 μm above to the injection site. In the ischemia group, the area of the skull overlying the M1 on the side subjected to ischemia was thinned, and an annular tube was fixed for laser illumination. The mice were allowed to recover for at least 2 weeks in their home cages. The Ca²⁺ fluorescent signals were recorded with a fiber photometry system (Thinker Tech, Nanjing) before (5 min), during (15 min), and after EA treatment (30 min) as previously described[90]. The beam of a 488 nm laser (OBIS 488LS; Coherent) was reflected by a dichroic mirror (MD498; Thorlabs), focused by a 10× objective lens (NA = 0.3; Olympus), and then coupled to an optical commutator (Doric Lenses). An optical fiber (230 mm OD, NA = 0.37, 1.5 mm length) guided the light between the commutator and the implanted optical fiber. The laser power at the tip of the optical fiber was adjusted to a low level (0.01–0.02 mW) to minimize bleaching. Ca²⁺ fluorescence was bandpass filtered (MF525-39, Thorlabs) and recorded by a photomultiplier tube (R3896, Hamamatsu). An amplifier (C7319, Hamamatsu) was used to convert the photomultiplier tube current (output) to voltage signals, which were further filtered through a low-pass filter (40 Hz cutoff; Brownlee 440). The analog voltage signals were digitalized at 500 Hz and recorded by a Power 1401 digitizer and Spike2 software (CED, Cambridge, UK). The data were exported into MATLAB for further analysis. We adjusted the baseline signal to the signal recorded at 5 min. We derived the fluorescence (ΔF/F) by calculating $(F-F_0)/F_0$, where $F_0$ was determined by averaging the baseline fluorescence signal during the initial 5 min.

## In vivo electrophysiological recording

In the experiment in which neuronal spikes in the NTS were recorded, as shown in Fig. 6e, f, 0.25 μl AAV2/9-CaMKIIα-ChR2-mCherry virus was injected into the M1, and AAV2/9-CaMKIIα−hM4Di-EGFP virus was injected into the PBN (AP: −5.2 mm, ML: 1.38 mm and DV: −3.4 mm). As shown in Fig. 6j, k, AAV2/1- CaMKIIα-Cre virus was injected into the M1, and AAV2/9-hSyn-Dio-mCherry virus was injected into the PBN. 14 d later, the mice were anesthetized with 1–2% isoflurane in an oxygen/air mixture and then fixed on a brain stereotactic apparatus (RWD, China). An incision was made in the middle of the scalp, and the skull was exposed. The target area was located, and an appropriate-sized skull window was drilled in the skull with a drill. A multichannel electrode was implanted into the NTS (AP: -6.48 mm, ML: 1.20 mm and DV: −4.04 mm), and an annular tube was fixed with dental cement for modeling. Then, the mice were allowed to recover for 7 d. The Plexon system (OmniPlex, Plexon, USA) was used for recording and further analysis. Neurons were recorded for 5 min to obtain baseline recordings before inhibition of PBN neurons by CNO intraperitoneal injection. 30 min after CNO injection, neuronal firing was recorded again for 5 min. After recording, the mice were sacrificed, and brain slices were prepared for further confirmation of virus expression or the recording site.

## Monosynaptic neuronal tracing

Neuronal tract tracing was performed on the mice as previously described[91]. As shown in Fig. 5g, h, the AAV2/1-hSyn-Cre-WPRE virus was stereotaxically injected into the M1, and AAV2/9-CAG-Dio-EGFP was stereotaxically injected into the PBN (AP: −5.2 mm, ML: 1.38 mm, DV: −3.4 mm). As shown in Fig. 6d, AAV2/1-hSyn-Cre-WPRE was injected into the PBN, and AAV2/9-CAG-Dio-EGFP was injected into the NTS. The Dio virus was used to assess the transmission of the Cre virus from the upstream brain regions to allow tracing of monosynaptic connections from the M1 to PBN or PBN to NTS. The mice were allowed to recover for 21 d to allow virus transduction. As shown in Sup Fig. 7, CTB-555 was stereotaxically injected into the PBN (AP: −5.2 mm, ML: 1.38 mm, DV: −3.4 mm) for retrograde tracing, and the mice were allowed to recover for -5 d. After recovery, the brain tissues of the virus-injected were collected for immunostaining.

## Recombinant rabies virus (RV)-based retrograde transsynaptic tracing

To explore the direct synaptic connections of the M1-PBN-NTS neural circuit, 0.3 µl rAAV-retro expressing Cre under the control of CaMKIIα promoter (rAAV-CaMKIIα-CRE-WPRE-hGH-pA, retro) was stereotaxically injected into the NTS (AP: −6.48 mm, ML: 1.20 mm and DV: −4.04 mm), and 0.3 µl of 1:1 mixture of AAV2/9-Ef1α-Dio-EYFP-TVA and AAV2/9-Ef1α-Dio-RVG was injected into the PBN (AP: −5.2 mm, ML: 1.38 mm and DV: −3.4 mm). After 21 d, the RV-EnvA-ΔG was injected into the PBN (Supplementary Fig. 9). The RV, whose *glycoprotein (G) gene* is deleted from the genome (RV-ΔG), cannot spread across synapses, but G complementation enables the transsynaptic spread of RV-ΔG to presynaptic neurons. The RV particles were packed with the fusion protein of the envelope protein (EnvA) of recombinant avian sarcoma virus, which can specifically recognize the EnvA receptor TVA to infect cells. With the aid of helper viruses, RV-EnvA-ΔG can be transmitted retrogradely to upstream neurons[48]. The mice were allowed to recover for 7 d and then were transcardially perfused with 0.9% saline and 4% PFA. Their tissues were fixed with 4% PFA overnight and dehydrated with 30% sucrose solution for three days. The brain tissues were embedded in OCT compound and sectioned with a freezing microtome at a thickness of 40 µm. These sections were imaged by confocal laser scanning microscopy.

## Data analysis and statistical tests

For animal experiments, normally distributed data were analyzed using one-way ANOVA with Tukey's post hoc test, unpaired *t* test, variance analysis of random block, and non-normally distributed data were analyzed by using Kruskal-Wallis with Tukey's post hoc test. Statistics with bilateral tests (*F* values or *P* values) are given in the figure legends. Unpaired Student *t* test was used for intergroup comparisons. Data analysis was performed by an experimenter blinded to the experimental conditions. SPSS (version 21.0) and GraphPad Prism 6 were used to conduct statistical analyses. All data are reported as the mean ± SEM. *N* indicates the number of biologically independent samples, mice per group. The significance level is denoted as $P < 0.05$.

## Reporting summary

Further information on research design is available in the Nature Portfolio Reporting Summary linked to this article.

## Data availability

All data supporting the findings of this study are included in the article and its supplementary information files and from the corresponding author. Source data are provided with this paper.

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

## Acknowledgements

We thank all the staff from the South China Research Center for Acupuncture and Moxibustion of Guangzhou University of Chinese Medicine and all the experimenters for their support to this study. All kinds of virus used in the experiments were purchased for the BrainVTA company. This work was supported, in part, by the National Key R&D Program of China (2019YFC1709100, 2019YFC1709102, to N.X.), the Key Program of the National Natural Science Foundation of China (81230088, to N.X.), the General Program of the National Natural Science Foundation of China (81774406, to N.X.), Youth Program of the National Natural Science Foundation of China (81904297, to L.W.), Key Laboratory Program of Universities in Guangdong province (XK2018001, to N.X.), Special project of "Lingnan modernization of traditional Chinese medicine" in 2019 Guangdong Provincial R & D Program (2020B1111100008, to N.X.), the Opening Operation of Key Laboratory of Acupuncture and Moxibustion of Traditional Chinese Medicine in Guangdong (2017B030314143, to N.X.), the Qi-Huang Scholar of National Traditional Chinese Medicine Leading Talents Support Program (2018, to N.X.), 'Elite Youth Education Program' of Guangzhou University of Chinese Medicine (QNYC20190106, to N.X.), Discipline Collaborative Innovation Team Program of Double First-class and High-level Universities for Guangzhou University of Chinese Medicine(2021XK01, to N.X.).

## Author contributions

N.G.X., L.L.Y., and L.W. designed all experiments. L.L.Y., Q.P.Y., Y.L., S.Q.Y., and S.Y. performed the experiments. Q.P.Y., Y.L., S.Q.Y., and S.Y. analyzed the data. Q.X., B.D., X.R.T., J.H.S., J.Y.L., J.S.W., and Y.L., L.Y., C.Z.T., and J.H.L. contributed to discussion. N.G.X., L.L.Y., Q.P.Y., Y.L., S.Q.Y., and L.W. wrote the manuscript.

## Competing interests

The authors declare that they have no competing interests. No competing financial or non-financial interest from the funders exist.

## Additional information

[1]South China Research Center for Acupuncture and Moxibustion, Medical College of Acu-Moxi and Rehabilitation, Guangzhou University of Chinese Medicine, Guangzhou 510006, China. [2]Department of Rehabilitation Medicine, The Third Affiliated Hospital, Sun Yat-sen University, Guangzhou 510630, China. [3]Department of Physiology, Institute of Acupuncture and Moxibustion, China Academy of Chinese Medical Sciences, 100700 Beijing, China. [4]Acupuncture Research Team, The Second Affiliated Hospital of Guangzhou University of Chinese Medicine, Guangzhou 510120, China. [5]These authors contributed equally: Lulu Yao, Qiuping Ye, Yun Liu, Shuqi Yao, Si Yuan. ✉e-mail: wanglin16@gzucm.edu.cn; ngxu8018@163.com

