## [Peer Review File · Nature Communications]

Electroacupuncture improves swallowing function in a post-stroke dysphagia mouse model by activating the motor cortex inputs to the nucleus tractus solitarii through the parabrachial nucleiREVIEWER COMMENTS

Reviewer #1 (Remarks to the Author):

The authors used a combination of nerve tracing and optogenetic methods in mice to identify the neural circuit activated during acupuncture therapy at pharyngeal acupoint CV23, which is located at the depression superior to the hyoid bone. This superficial electrical stimulation method is widely used in China to treat post-stroke dysphagia (PSD); however, the underlying mechanisms are largely unknown. Thus, the authors created a mouse model of PSD to specifically address this knowledge gap.

The authors are commended for conducting this comprehensive pre-clinical investigation of dysphagia in PSD. This work appears to have high translational importance, but it is very difficult to follow and grasp in its current form. Editing the entire manuscript for English language is essential for improved readability and clarity of intent. Moreover, adding an "Experimental Methods" table that lists each experiment and summarizes relevant details (e.g., number of mice, injection location, purpose/research question/hypothesis, and key findings) would add tremendous clarity to this paper, allowing readers to more quickly follow along with and understand this complex work.

Additional details/information is requested in the following sections:

1. Introduction: Please include details of CV23 stimulation treatment effects in PSD patients. What "dysphagia outcome measures" are specifically improved by this acupuncture therapy approach?

2. Methods: Some "invasive" experiments (or portions thereof) appear to be conducted without sedation/anesthesia. For example, mice are tail vein-injected with rose bengal and then anesthetized for stereotaxic surgery. In this case, why wasn't the mouse anesthetized for the tail vein injection, too? Also, EMG recordings appear to be done without sedation/anesthesia, even though mice are fixed in supine position and wire electrodes inserted into the neck muscles, and a PE tube placed "in the mouse's hard palate." Why does this invasive approach not require sedation/anesthesia? Was the PE tube somehow attached to the surface of the hard palate or surgically implanted?

3. Methods: What is the rationale for the stimulation parameters (frequency, intensity, duration, timing)?

4. Results: What is meant by "improved swallowing function"? What specific swallow-related outcome measures were improved?

5. Discussion: Please include an interpretation of how the improved swallow-related outcome measures in PSD mice following C23 stimulation may translate to human PSD patients?

Reviewer #2 (Remarks to the Author):

This is a very comprehensive and potentially important paper on the neurophysiology of swallowing. It represents a significant amount of work. I am not qualified to review the efficacy of all the viral vector based interventions so I will leave to other reviewers to determine their validity. I will focus on the evaluation of swallowing function and the interpretation of the results regarding electrostimulation of mylohyoid.

The novelty of this paper lies not so much in demonstrating the effectiveness of electrostimulation of suprahyoid muscle in improving function after stroke but in postulating a neurophysiological mechanism for this through very detailed anatomical and functional tracing of neurological circuits from the primary motor cortex to the brainstem. This is a huge amount of work. I would like it clarified how many mice received each treatment, and very specifically I would like to know how many different treatment groups were there for the mice with stroke and electrostimulation? It is difficult to

keep track of all the different procedures that were performed and to evaluate the results it is important to know how many experimental cohorts there were.

Looking at the sample sizes in the figure legends, not all group sizes were equivalent and some groups seem very small. Have the authors considered the possibility of sampling issues and study power in their design?

It is also unclear to me what EMGS were in fact measured. The authors merely describe placing electrodes in "neck muscles". This to me would mean the infrahyoid muscles, not the mylohyoid.

Could the authors provide clarification on electrode placement?

The authors also state they implanted their electrical stimulation device ".5 cm into the mouse". Did the authors perform any dissection to confirm electrode location? At the location authors describe in the mouse, many muscles are dorso-ventrally superimposed (mylohyoid, geniohyoid, hyoglossus).

How did the authors confirm the position of the stimulating electrode within the muscle layers?

Similarly, the authors argue that this point is the same as one use in human acupuncture. However, the comparative anatomy of the hyoid region between rodents and humans makes a simple like for like comparison inappropriate. In humans, the mylohyoid is thinner (particularly posteriorly close to the hyoid), and in general the floor of the mouth is mediolaterally broader and cranio caudally

compressed. Thus whether or not the electrode placed in the mouse is stimulating the same muscles as an electrode placed at CV23 in humans cannot be determined without anatomical comparison.

Similarly, a depth of .5 cm in mouse represents a significant proportion of the musculature. Would an acupuncture needle reach an equivalent depth in humans?

In general I think the acupuncture framing detracts from the basic science/neurophysiological significance of this work.

Finally, the paper requires significant editing. There are many sentences that do not end properly, or are unclear. The methods section is difficult to follow. The problem and the hypothesis are not clearly stated (the last paragraph of the introduction is in fact a summary of the results).

There is a lot of potentially very good and important work in this paper, but it is obscured by poor structure, inadequate framing, and incomplete reporting of experimental conditions.

Reviewer #3 (Remarks to the Author):

In this study, the authors tried to reveal a neuroanatomical basis for electroacupuncture (EA) to promote swallowing in mice with post-stroke dysphagia (PSD). Using retrograde labeling, they first revealed a descending neural circuit from M1, to parabrachial nuclei (PBN), then to the nucleus of solitary tract (NTS), and finally to the mylohyoid muscle. They then found that EA at the CV23 acupoint within the mylohyoid muscle can increase M1 activity in the contralateral intact side in PSD mice, and this activation can promote swallowing via the M1-PBN-NTS circuit. Silencing this circuit led to loss of EA effects. In overall, the finding is interesting. However, a number of issues needs to be addressed.

Based on a modest retrograde labeling of M1 neurons by PBN neurons receiving inputs from NTS, the authors claimed that PBN neurons receiving inputs from M1 are different from those receiving inputs from NTS, indicating the presence of some unknown intermediate steps. However, it is fully possible that only a very small subset of PBN neurons with M1 inputs are involved in swallowing. Thus, it does not rule out the possibility that those "few" connected M1-PBN-NTS neurons are involved in swallowing.

There were no studies or discussions on how stimulation in the CV23 acupoint could lead to M1 activation. C-Fos induction in candidate neural circuits should be examined.

The details of statistical analyses in Figure legends are missing

Figure S5: the "representative" c-Fos images do not match the quantitative data.

Dear reviewers,

We appreciate the critical and thoughtful comments, questions and suggestions from the reviewers. Accordingly, we have performed additional experiments and extended relevant discussions to address the concerns raised, and intensively polished the English expression to improve the readability and clarity of the manuscript. In addition, we have reformatted the manuscript to meet the journal's requirement and added a table to clarify the experimental details in each figure.

Please find our detailed point-by-point responses below. We hope that the revised manuscript could meet the standard of your journal for publication. Thank you very much.

Sincerely,

Prof. Nenggui, Xu

Point-by point responses (MS ID: NCOMMS-21-36223)

Reviewer #1 (Remarks to the Author):

1. The authors are commended for conducting this comprehensive pre-clinical investigation of dysphagia in PSD. This work appears to have high translational importance, but it is very difficult to follow and grasp in its current form. Editing the entire manuscript for English language is essential for improved readability and clarity of intent. Moreover, adding an "Experimental Methods" table that lists each experiment and summarizes relevant details (e.g., number of mice, injection location, purpose/research question/hypothesis, and key findings) would add tremendous clarity to this paper, allowing readers to more quickly follow along with and understand this complex work.

Response: Thanks for your suggestions. The English language and expression of the entire manuscript have been extensively improved by native speakers. A table listing the key details of each experiment including the purpose, virus injection, and numbers of mice, has been added on the revised manuscript (Pages 47-49).

2. Additional details/information is requested in the following sections:

(1). Introduction: Please include details of CV23 stimulation treatment effects in PSD patients. What "dysphagia outcome measures" are specifically improved by this acupuncture therapy approach?

Response: According to the results obtained from the clinical trials, the methods have been included for evaluating the acupuncture therapy in PSD patients as following: (1) the Videofluoroscopic Swallowing Study (as a standard and visualized evaluation of the swallowing function by measuring the Inter-Swallow Interval, Lick-Swallow Ratio, Pharyngeal Transit Time, Esophageal Transit Time and the size of food); (2) Fiberoptic Endoscopic Evaluation of Swallowing (FEES, as a useful supplementary tool for evaluation of the swallowing function by observing the structure of the pharynx and larynx with different textures and sizes of food and liquid); (3) Standardized Swallowing Assessment (SSA, as a psychometric evaluation of the swallowing process by measuring the conscious, body control, breathing, oral closure, laryngeal function pharyngeal reflex and spontaneous cough); (4) Kubota Water Swallowing Test (WST, as a simple routine screening test by observing the frequency and time of drinking water and the cough); (5) pharyngeal surface electromyography (PsEMG, as an evaluation of the characteristics of external musculature by quantifying the root mean square, integrated electromyography and averaged electromyography) These measurements and incidence of dysphagia complications was lowered by the treatment (PMID: 30945493, 34698463). This has been added into the Introduction Section (Pages 3-4).

(2). Methods: Some "invasive" experiments (or portions thereof) appear to be conducted without sedation/anesthesia. For example, mice are tail vein-injected with rose bengal and then anesthetized for stereotaxic surgery. In this case, why wasn't the mouse anesthetized for the tail vein injection, too? Also, EMG recordings appear to be done without sedation/anesthesia, even though mice are

fixed in supine position and wire electrodes inserted into the neck muscles, and a PE tube placed "in the mouses's hard palate." Why does this invasive approach not require sedation/anesthesia? Was the PE tube somehow attached to the surface of the hard palate or surgically implanted?

Response: Thanks for pointing out these issues. Apologize for the confusion caused by our written mistakes. For the tail vein injection experiment, mice were injected with rose bengal after being anesthetized (Pages 16).

For recording the EMG responses induced by water, the reasons for not subjecting mice to anesthesia are following: (1) Mice were body-fixed by an adaptor, and the electrodes inserted into mylohyoid could be recorded without anesthesia. The schematic diagram was showed in the Sup.Fig.2a. (2) Responses were too few during anesthesia to be analyzed. (3) The tube was placed roughly at surface of the tongue with no leading to the obvious struggling.

(3). Methods: What is the rationale for the stimulation parameters (frequency, intensity, duration, timing)?

Response: Thanks for your concerns about the stimulation parameters. Stimulation with **2Hz/15min** was applied in the present study based on the evidence from our clinical study and previous reports. Firstly, our unpublished data in human volunteers suggested that stimulation with **2Hz/15min** showed the greatest efficacy compared to 2Hz with, 5min, 15min, 30min, and 100Hz with 5min, 15min, 30min. Moreover, the published data (PMID: 29473351) demonstrated the efficacy of electroacupuncture with low frequency (**2Hz**) is better than that with high frequency (100Hz) in PSD patients. Besides that, electroacupuncture with **2Hz** is mostly widely used for traditional treatment in the clinic. Secondly, the effect of different frequencies including 2Hz/50Hz/100Hz of electroacupuncture at CV23 in mice has been explored, the study (PMID: 32519209) suggested the stimulation with **2Hz** represented larger improvements in swallowing function, compared to that with 50Hz/100Hz stimulation. In terms of intensity, in the recent paper published in Neuron (PMID: 32791039), showed that **1.0 mA** intensity stimulation at another widely used acupoint, ST36, located at hindlimb and in proximity of the common peroneal and tibial branches of the sciatic nerve, produced anti-inflammatory effect. In addition, the stimulation with **1.0 mA** intensity could lead to the mice showing visible muscle contraction near the acupoint during stimulation, which is an important signal for estimating the effectiveness for this treatment. For the consideration of timing, the EA treatment was given **24h after stroke**. At this point, significant EA mediated effect to swallowing function and blood flow was significantly occurred (PMID: 32519209, and 33061953). Therefore, EA stimulation with 2Hz/15min, 1.0mA was selected to treat in mice 24h post stroke induction.

(4). Results: What is meant by "improved swallowing function"? What specific swallow-related outcome measures were improved?

Response: The EMG response recording and water consumption test were used to assess the

swallowing function. The EMG recording, mimicking the pharyngeal surface electromyography in clinics, detects the neuromuscular function by measuring the muscle response to a nerve's stimulation of the muscle, and it is extensively used to assess the muscles' activity in the previous studies (PMID: 26436986, 16879438, 15655687). The AUC (Area under curve) reflects the number and diameter of the muscle fibers activated during the swallowing process. As the muscle is contracted more forcefully, more and more muscle fibers are activated, and the larger AUC responses are observed during recording. Thus, the EMG responses obtained from the mylohyoid could reflect the ability of its contraction during swallowing process. While the water consumption test, mimicking Kubota Water Swallowing Test used in the clinical practice, assessing the ability of water intake, the smaller water consumption was supposed to be positively associated with the impaired swallowing function.

(5). Discussion: Please include an interpretation of how the improved swallow-related outcome measures in PSD mice following C23 stimulation may translate to human PSD patients?

Response: Thanks for your suggestion. As mentioned in the previous answer (4), the water consumption test and EMG recording at mylohyoid were used in the experiments to measure the swallow-related outcome. The EMG recording is used to detect the neuromuscular function by measuring the muscle response to a nerve's stimulation of the muscle. For the mylohyoid is involved in the initiation of swallow and functioning as the laryngeal elevators puling the larynx upward. Thus, the EMG responses to some extents are reflecting the swallowing function. This EMG recording method has been extensively applicated for a long time to measuring of swallowing function (PMID: 9503343, 4029536, 14652082). For the water consumption test, it is complementary method to mimic the Kubota Water Swallowing Test evaluating the ability to drink water. While in the case of clinical practice, VFSS and FEES are the gold standard measurements of the swallow function, we are trying to establish the VFSS and FEES to assess the swallowing function and the effect of EA-CV23 in the mice as reported by the Terasa E. Lever's lab. In general, this study revealed that the swallowing function could be improved by EA-CV23 in PSD mice, and the brain regions involving regulation of the swallowing function could be modulated by EA-CV23. This interpretation has been included in the Discussion in the revised version (Pages 13)

Reviewer #2 (Remarks to the Author):

1. The novelty of this paper lies not so much in demonstrating the effectiveness of electrostimulation of suprahyoid muscle in improving function after stroke but in postulating a neurophysiological mechanism for this through very detailed anatomical and functional tracing of neurological circuits from the primary motor cortex to the brainstem. This is a huge amount of work. I would like it clarified how many mice received each treatment, and very specifically I would like to know how many different treatment groups were there for the mice with stroke and electrostimulation? It is difficult to keep track of all the different procedures that were performed and to evaluate the results it is important to know how many experimental cohorts there were.

Response: Thanks so much for your appreciation of our work on the neurophysiological mechanism. In this study, we are trying to explore the mechanism how the EA-CV23 regulates the swallowing function from the perspective of the neural circuit. In the study, it is our limitation that we have not provided the more direct results showing the effectiveness of this stimulation in mice. We are trying to develop the transoral endoscopy and videofluoroscopic Swallowing Study, as reported by Teresa E. Lever's lab. Our preliminary results obtained by transoral endoscopy showed the asymmetric and incomplete closure in the PSD, while attenuated by the EA-CV23 treatment (See below)

The pharyngeal structure under transoral endoscopy. (A)The symmetrical movement of vocal cords were visible. (B-C) The vocal cord movement was asymmetric and incomplete closure in the PSD, and the vocal cord movement was restored after treatment by EA-CV23.

The experimental list including purpose/experiment/numbers of mice has been added to clarify the results.

2. Looking at the sample sizes in the figure legends, not all group sizes were equivalent and some groups seem very small. Have the authors considered the possibility of sampling issues and study power in their design?

Response: Thanks so much for your points. The sample size has been increased in those small N groups in revised version (Fig.6f-h). For each experiment, the size in the group is equivalent, while the sample size is different among experiments.

Your comments on sampling issue and study power are key and important in the clinical trials. In our study on the animal experiments, the sampling issue and study power has not been conducted. The reasons are follows: (1) It is hard to design the study power before experiments, for these animal

experiments are the explorative studies without no published data/results. (2) In addition, the properties from animal samples represented higher homogeneous, comparing to that from the clinics. (3) Moreover, most previous published paper on the animal experiments included the similar sample size (PMID: 34646018, 32791039, 31517050).

3. It is also unclear to me what EMGS were in fact measured. The authors merely describe placing electrodes in "neck muscles". This to me would mean the infrahyoid muscles, not the mylohyoid. Could the authors provide clarification on electrode placement?

Response: The EMG recording could detect the neuromuscular function by measuring the muscle response to a nerve's stimulation of the muscle. The AUC (Area under curve) analyzed reflects the number and diameter of the muscle fibers activated during the swallowing process. As the muscle is contracted more forcefully, more and more muscle fibers are activated. The location placing the electrodes has been clarified by injecting the dye to label the electrode position (Sup.Fig.1a). It was shown that the mylohyoid muscle was stained. We have made a clearer interpretation of the electrode placement in the Methods (Pages 15).

4. The authors also state they implanted their electrical stimulation device ".5 cm into the mouse". Did the authors perform any dissection to confirm electrode location? At the location authors describe in the mouse, many muscles are dorso-ventrally superimposed (mylohyoid, geniohyoid, hyoglossus). How did the authors confirm the position of the stimulating electrode within the muscle layers?

Response: As you noted, there are many muscles superimposed in mice, including the digastric, mylohyoid, geniohyoid, and hyoglossus. However, there is no specific marker distinguishing these different muscles, thus the location was confirmed by the dye. The dye was injected at the location same as the site acupuncture needle inserting, as shown in Sup.Fig.1a. We have performed the dissection of these musculatures following the reference and confirmed the muscles.

5. Similarly, the authors argue that this point is the same as one use in human acupuncture. However, the comparative anatomy of the hyoid region between rodents and humans makes a simple like for like comparison inappropriate. In humans, the mylohyoid is thinner (particularly posteriorly close to the hyoid), and in general the floor of the mouth is mediolaterally broader and cranio caudally compressed. Thus whether or not the electrode placed in the mouse is stimulating the same muscles as an electrode placed at CV23 in humans cannot be determined without anatomical comparison. Similarly, a depth of .5 cm in mouse represents a significant proportion of the musculature. Would an acupuncture needle reach an equivalent depth in humans?

Response: Thanks for your advice. We agree that there are differences in anatomy of the hyoid regions between human body and mice, and could not be simply compared. It is the shortcoming of our study that the direct evidence on the specific location of the needle/electrode has not been verified in the clinics, although the EA at CV23 acupoint was defined to locate at mylohyoid (1.

Shao, S., Acupoint anatomy.2016; 2. Lim, S., WHO Standard Acupuncture Point Locations.2010). Even so, the depth of the needle insertion in the PSD patient with the videofluoroscopy was observed and estimated as about 1.25cm (The depth is dependent on individual proportions of each body, including the height, weight, and the thickness of the tissue, ect.) (See below). The depth of position needle insertion in humans was supposed to be enough to stimulate the mylohyoid, Moreover, for the safety in the clinics, the needle would not reach the geniohyoid muscle, where located below the mylohyoid and close to the trachea. Therefore, the depth of .5cm in mouse is suggested to be comparable to that in humans.

The image showing the position of acupuncture needle inserting with videofluorocopy

It's worth noting that these mylohyoid, geniohyoid, hyoglossus muscles are superimposed as you mentioned, in addition to the mylohyoid stimulated/ recorded during the EA stimulation or EMG recording, the geniohyoid or hyoglossus are likely affected as well. From the perspective of the modern medical theory, we hypothesized that it is possible that it would take the similar effect by stimulating the muscles nearby, including the mylohyoid, geniohyoid, hyoglossus. For all these musculatures are involved in the swallowing, and innervated by hypoglossal or trigeminal nerves. These considerations have been added into the new version (Pages 13).

6. In general I think the acupuncture framing detracts from the basic science/ neurophysiological significance of this work.

Response: Thanks so much for your appreciating the neurophysiology of this work. The efficacy of acupuncture treatment has been widely accepted in the clinical practice (PMID: 24595780, 32213509). However, its underlying mechanism is largely unknown, and difficult to explore. Recently, some high-qualified paper on the mechanism of acupuncture therapy of the pain have published. Liu et al revealed the neuroanatomical basis for the selectivity and specificity of acupoints in driving autonomic pathways for anti-inflammation (PMID: 34646018). In this study, we showed that electrostimulation of the CV23 acupoint, regulated the swallowing process via activation the M1-PBN-NTS neural circuits. This mechanism might also be involved in the other peripheral stimulation for the dysphagia treatment, including pharyngeal electric stimulation (PES) and laryngopharyngeal neuromuscular electrical stimulation (NMES). The therapy might activate the similar pathway of nerve stimulation or signal conduction. In general, we, together with those

interested in the basic mechanism of acupuncture, are trying to uncover the veil of this old traditional medical therapy from the modern medical perspective. Moreover, compared to these PES, NMES methods, EA stimulation, with its properties of safety, easiness to operate, convenience and inexpensiveness, might be more welcomed as dysphagia therapy.

7. Finally, the paper requires significant editing. There are many sentences that do not end properly, or are unclear. The methods section is difficult to follow. The problem and the hypothesis are not clearly stated (the last paragraph of the introduction is in fact a summary of the results).

Response: Thanks so much for your suggestion. We have made a thorough edition on our writing.

8. There is a lot of potentially very good and important work in this paper, but it is obscured by poor structure, inadequate framing, and incomplete reporting of experimental conditions.

Response: Thanks for the suggestion. We have made a great amount of editing to better convey the findings in our study in the revised version.

Reviewer #3 (Remarks to the Author):

1. Based on a modest retrograde labeling of M1 neurons by PBN neurons receiving inputs from NTS, the authors claimed that PBN neurons receiving inputs from M1 are different from those receiving inputs from NTS, indicating the presence of some unknown intermediate steps. However, it is fully possible that only a very small subset of PBN neurons with M1 inputs are involved in swallowing. Thus, it does not rule out the possibility that those “few” connected M1-PBN-NTS neurons are involved in swallowing.

Response: Thanks for your points. The role of PBN neurons with M1 inputs has been explored and presented in the revised version. The AAV2/9-CaMKII α -ChR2-mCherry virus was injected into M1, while optical fiber was implanted into PBN region to specifically to activate the axon terminals in PBN from the M1 projection. This activation of M1-PBN neural circuit induced a significant swallowing response (Fig.5f). Based on the strong structural monosynaptic connections between M1 and PBN (Fig.4b), it is not likely that there is only small subset of PBN neurons with M1 inputs are involved in the swallowing process.

To further clarify the contribution of M1-PBN-NTS circuit directly, AAV2/1-CaMKII α -Cre virus into M1 and AAV2/9-EF1a-DIO-hChR2-EGFP into PBN, and the optical fiber was implanted into the NTS. Although the optogenetic activation terminals from PBN receiving the M1 input can induce significant EMG responses, activation of direct connected M1-PBN-NTS neurons is much smaller comparing to that induced by the activation of M1 neurons (Fig.6j). This suggested the modulation of swallowing function is partly associated with the direct connected M1-PBN-NTS neurons, while the M1-PBN-NTS neural circuit involved different clusters of neurons within PBN played the major role in the swallowing process (Fig.7).

2. There were no studies or discussions on how stimulation in the CV23 acupoint could lead to M1 activation. C-Fos induction in candidate neural circuits should be examined.

Response: Thanks so much for your suggestion. The candidate neural circuits have been explored and shown as Sup Fig6. In addition to the M1, PBN and NTS, the PVH, hypothalamus, LSV and thalamus were observed with increased expression of c-Fos. We hypothesized that this stimulation may involve the excitation of sensory nerves nearby, such as hypoglossal nerve and trigeminal nerve, and these nerves could transmit the sensory swallowing information to sensory cortex and then activate the M1 via the spinal cord and cranial nerves to brainstem neuron network. The increased expression of c-Fos in primary sensory cortex observed after the EA-CV23 supported this hypothesis (Sup Fig.6). The afferent pathway of how stimulation at the CV23 acupoint leads to M1 activation requires much exploration to clarify in the further study. This has been included into discussion in the revised version (Pages 14).

3. The details of statistical analyses in Figure legends are missing

Response: The details of statistical analysis have been added into Figure legends as suggested.

4. Figure S5: the “representative” c-Fos images does not match the quantitative data.

Response: This mistake has been corrected in the revised version.

REVIEWER COMMENTS

Reviewer #1 (Remarks to the Author):

Although the authors have improved the use of English language throughout the manuscript, there are still numerous instances of agrammatism that require substantial editing by a native English language speaker for improved flow and/or clarity. A few representative examples are listed below (but these are but a very few of the numerous examples that could have been selected from the entire manuscript):

Page 2, line 28: that could regulate the (delete "the") mylohyoid activity to regulate the (delete "the") swallowing function in mice.

Page 3, line 49: "...stimulation, has been utilized in the (delete "the") clinical practice for treatment of..."

Page 3, line 58: ".... In the (delete "the") clinical practice, ..."

Page 4, line 68: "...closure, laryngeal functon(add comma here) pharyngeal reflex and spontaneous cough) and the..."

Page 4, lines 69-70: "...(WST, as a simple routine screening test by observing the frequency and time of drinking water and the cough)...". What is meant by "and the cough" here?

Page 4, lines 73-74: "Despite its widely (should be "wide") application..."

Page 4, line 76: "...EA-CV23 improves the (delete "the") swallowing function in the PSD mouse model."

Page 4, line 87: "...the brain stem, is also thought to be important for (add "the" here) swallowing reflex."

Page 4, line 93-94: "...however, there are (change to "is") no direct evidence illustrating how these brain regions jointly participate in the regulation of the (delete "the") swallowing function."

Page 5, lines 117-118: "Then we investigated whether this connection could participate the EA-CV23 improving swallowing process." This sentence does not make sense.

Page 7, lines 137-139: "To investigate the swallowing difficulty could be induced by M1 injury, the ischemia mice was conducted by intraperitoneally injected with rose Bengal and subjected to laser irradiation of M1 region, or PFC region as a control..." This sentence as written is very difficult to read and comprehend.

Page 7, 157-159: "We next to determine whether EA-CV23 modulated the neuronal activity of contralateral M1 L5 involving the improvement of the swallowing function for PSD treatment." This sentence as written is very difficult to read and comprehend.

Page 16, lines 392-393: "After recovery, mice were given with 473 392 nm light..." What is meant by "given with" here? Do you mean "treated with"?

Page 16, lines 406-407: "...the balloon was fixed on the pharyngeal and the EMG recording electrodes were inserted..." This sentence is incomplete and therefore does not make sense. The methodology is incompletely described here. Were the mice still fully awake during the balloon pressure detection experiment? How was the balloon position verified? Did the balloon mechanically obstruct breathing by compressing the soft palate?

Page 16, lines 409-410: “The pressure of the balloon and EMG responses were detected simultaneously with (should be “during”) blue light stimulation.”

In addition, several areas throughout the manuscript require additional clarification before the reported results and discussion can be fully assessed/interpreted by the reader. Several examples are listed below:

Page 2, lines 32-33: “... and the EA-CV23’s 32 improvement of swallowing function for PSD treatment.” A brief description of dysphagia in this model is needed in the abstract (what evidence of dysphagia exists for this model?), as well as a summary of what aspects of dysphagia were specifically improved by EA-CV23 treatment. This important detail should be expanded upon in the methods, results, and particularly the discussion sections of the manuscript for added impact.

Page 3, line 62: “lick-swallow ratio” is not a standard measure of swallowing function in clinical practice – it’s a measure used in animal studies only.

Page 4, lines 70-73: “...pharyngeal surface electromyography (PsEMG, as an evaluation of the characteristics of external musculature by quantifying the root mean square, integrated electromyography and averaged electromyography) in the clinical practice...” What is meant by “external musculature”? What exactly is being measured clinically via EMG that translates to this animal study?

Page 5, lines 97-100: “In the study, we found that optogenetic activation of the M1 L5 excitatory neurons could elicit frequency-and intensity-dependent EMG responses in mylohyoid, and unilateral photochemical ischemia of M1 was sufficient to produce dysphagia, establishing the PSD mouse model.” Separate these two “concepts” into different sentences for emphasis of importance. This is the first mention of optogenetics – explain why it was used here for clarification and added impact. Also briefly describe the PSD model with appropriate references – why was this model specifically chosen for this work?

Page 6, lines 115-116: “...the EGPF-positive neurons were detected in the M1 Layer 5 (L5) 96h instead of 48h or 72h after injection...” Please clarify why “instead of 48h or 72h after injection” is important information to know here. Why 48h and 72h, specifically?

Page 6, lines 126-127: “Optogenetic activation of M1 neurons induced the increase in the mylohyoid activity and pharyngeal pressure...” How was pharyngeal pressure measured here? A brief summary would be helpful.

Pages 15-16, lines 386-387: “... and a tube (0.85*0.42 mm PE tube) for water delivery was placed in the mouse’s hard palate.” Please clarify “placed in” – this sounds like a surgical procedure. Do you mean “positioned against” rather than “placed in”?

Page 16, Line 438: “EA procedure in mice”. Please clarify “EA” here in addition to using the acronym.

Page 17, lines 442-443: “...at about a 0.5cm depth into CV23 acupoint, which located at the mylohyoid.” Do you mean “at the presumed mylohyoid location” based on previous work (explain or cite here)? The details are included in the authors “rebuttal” but not incorporated into the manuscript.

Page 17, lines 438-445: “Electrical stimulation of 2 Hz with 1 mA strength of intermittent pulses for 15 min was performed...” Please clarify here why these parameters were specifically used. The details are included in the authors “rebuttal” but not incorporated into the manuscript.

Page 17, lines 445-447: “. A sham EA control treatment followed the same procedure but used a non-electrical wooden toothpick instead of the stainless-steel needle...” Why not simply unplug the

stainless-steel electrode from the stimulator for the control condition instead of using an entirely different electrode (different material, size, etc.) for the control condition?

Page 17, lines 449-456: Methodological details of the water swallow test are incomplete; therefore, replication efforts based on the available details would be impossible. What type of bottles and spouts were used, what volume of water did they hold, what type of water was used (pH-adjusted filtered tap water or DI water), was the water room temperature, etc. Did the mice drink in their home cage or were they tested individually in separate cages? (Also clarify if mice were single housed for the entire experiment – this detail is missing). Were the bottles leak-tested to determine the amount of water that spontaneously leaks from the bottles during the typical water swallow test duration (without a mouse drinking from the spout)?

Reviewer #2 (Remarks to the Author):

I am broadly satisfied with the way in which the authors have addressed my concerns as a reviewer but note that other reviewers have raised more specialized concerns about the techniques used I am not able to evaluate. However as far as the concerns I raised about the paper I am satisfied for it to be published.

Reviewer #3 (Remarks to the Author):

The authors have addressed most issues. However, some issues remain:

- 1) For c-Fos induction shown in Sup. Figure 6: there is no control Fos with EA.
- 2) A new issue emerges. The whole studies tried to suggest information flow from M1 to PBN and finally to NTS in driving swallowing. However, the tools used appeared not to rule out a reverse direction: NTS-PBN-M1. Could AAV2/9 injection in M1 retrogradely label PBN neurons projecting to M1, and could the reported monosynaptic connections between M1-PBN be due to antidromic activation of PBN terminals in M1 and back to soma? AAV2/1 anterograde labeling is known to contain some retrograde activity. As such AAV2/1 injection in M1 may also retrogradely label PBN neurons. AAV2/1 injection in PBN could also retrogradely label NTS neurons. The strongest evidence for M1-PBN-NTS direction is the silencing of NTS firing in response to M1 stimulation following PBN silencing, although some forms of reciprocal connections between M1-projecting PBN neurons and NTS neurons may need to be ruled out. The authors may need to address some of these concerns and then provide some discussions if they cannot fully address this possibility.

Point-by-point responses (MS ID: NCOMMS-21-36223A)

Reviewer #1 (Remarks to the Author):

Although the authors have improved the use of English language throughout the manuscript, there are still numerous instances of agrammatism that require substantial editing by a native English language speaker for improved flow and/or clarity. A few representative examples are listed below (but these are but a very few of the numerous examples that could have been selected from the entire manuscript):

Response: Thanks for your constructive suggestions, which is valuable for improving the language of the manuscript. We have carefully scrutinized the entire manuscript and asked a native and professional English language speaker for help to edit and make the corresponding revisions of the agrammatisms.

1. Page 2, line 28: that could regulate the (delete "the") mylohyoid activity to regulate the (delete "the") swallowing function in mice.

Response: These two “the” have been deleted. (Pages 2, Line 27-28)

2. Page 3, line 49: “...stimulation, has been utilized in the (delete "the") clinical practice for treatment of...”

Response: This has been deleted. (Pages 3, Line 51)

3. Page 3, line 58: “.... In the (delete "the") clinical practice, ...”

Response: This has been deleted. (Pages 3, Line 58)

4. Page 4, line 68: “...closure, laryngeal functon(add comma here) pharyngeal reflex and spontaneous cough) and the...”.

Response: The comma here has been added. (Pages 3, Line 64)

5. Page 4, lines 69-70: “...(WST, as a simple routine screening test by observing the frequency and time of drinking water and the cough)...”. What is meant by “and the cough” here?

Response: The sentence here has been revised to improve its clarity. The cough here indicated the presence of cough after swallow. (Pages 3, Line 65-66)

6. Page 4, lines 73-74: “Despite its widely (should be “wide”) application...”.

Response: This has been modified. (Pages 3, Line 68)

7. Page 4, line 76: “...EA-CV23 improves the (delete "the) swallowing function in the PSD mouse model.”

Response: This has been deleted. (Pages 4, Line 70-71)

8. Page 4, line 87: “...the brain stem, is also thought to be important for (add “the” here) swallowing reflex.”

Response: It has been added. (Pages 4, Line 81-82)

9. Page 4, line 93-94: “...however, there are (change to “is”) no direct evidence illustrating how these brain regions jointly participate in the regulation of the (delete "the) swallowing function.”

Response: This sentence has been modified as suggested. (Pages 4, Line 87-88)

10. Page 5, lines 117-118: “Then we investigated whether this connection could participate the EA-CV23 improving swallowing process.” This sentence does not make sense.

Response: This sentence has been revised as “Then we investigated whether the function of this connection was associated with the regulation of swallowing activity”. (Pages 6, Line 115-116)

11. Page 7, lines 137-139: “To investigate the swallowing difficulty could be induced by M1 injury, the ischemia mice was conducted by intraperitoneally injected with rose Bengal and subjected to laser irradiation of M1 region, or PFC region as a control...” This sentence as written is very difficult to read and comprehend.

Response: To better convey the information, this sentence has been rewritten as “To investigate whether swallowing difficulty could be induced by M1 injury, we firstly established the focal ischemia in M1 by intraperitoneal injection with Rose Bengal and laser irradiation in the M1 region”. (Pages 7, Line 135-137)

12. Page 7, 157-159: “We next to determine whether EA-CV23 modulated the neuronal activity of contralateral M1 L5 involving the improvement of the swallowing function for PSD treatment.” This sentence as written is very difficult to read and comprehend.

Response: To clarify this sentence, we have made the revision as follows: “We next determined whether EA-CV23 improved swallowing function by modulating the neuronal activity in contralateral M1 L5”. (Pages 7, Line 158-159)

13. Page 16, lines 392-393: “After recovery, mice were given with 473 392 nm light...” What is meant by “given with” here? Do you mean “treated with”?

Response: As you mentioned, it meant “treated with” here, and we have made the revision accordingly. (Pages 18, Line 442)

14. Page 16, lines 406-407: “...the balloon was fixed on the pharyngeal and the EMG recording electrodes were inserted...” This sentence is incomplete and therefore does not make sense. The methodology is incompletely described here. Were the mice still fully awake during the balloon pressure detection experiment? How was the balloon position verified? Did the balloon mechanically obstruct breathing by compressing the soft palate?

Response: During the Balloon Pressure Detection experiment, the mice were fixed in the supine position and kept awake.

As the issue of balloon position you concerned, firstly, the position of balloon is near the location inserting with the tube for water supply; secondly, the balloon against within the root of tongue could detect the changes of pressure during swallowing movement; thirdly, if the balloon away from the indicated position, the pressure signal would not be detected.

During the experiment, the breath was not affected by the balloon for its small diameter of 3mm, and there is room enough for breathing. Moreover, the signals were detected when the experimental mice stayed peaceful without apparent struggling.

These details have been included into the revised text. (Pages 18-19, Line 456-462)

15. Page 16, lines 409-410: “The pressure of the balloon and EMG responses were detected simultaneously with (should be “during”) blue light stimulation.”

Response: This part has been modified as instructed. (Pages 19, Line 466)

In addition, several areas throughout the manuscript require additional clarification before the reported results and discussion can be fully assessed/interpreted by the reader. Several examples are listed below:

Response: Thank you for your reminding. According to your nice suggestions, we have made additional clarifications, which contributed a lot to improve the quality of the article.

16. Page 2, lines 32-33: "... and the EA-CV23's 32 improvement of swallowing function for PSD treatment." A brief description of dysphagia in this model is needed in the abstract (what evidence of dysphagia exists for this model?), as well as a summary of what aspects of dysphagia were specifically improved by EA-CV23 treatment. This important detail should be expanded upon in the methods, results, and particularly the discussion sections of the manuscript for added impact.

Response: Thank you for your valuable comments to improve the impact of our study. Our previous studies reporting this model have been published (PMID: 33061953, 32519209). These papers demonstrated the impaired EMG responses in the mylohyoid and decreased water consumption in PSD mice, and the pathological dysfunction could be improved by EA-CV23 treatment. (Abstract, Pages 2, Line 29-31; Results, Pages 6, Line 140-153)

As mentioned in the introduction (Pages 3, Line 58-68), all FEES, VFSS, SSA, WST and PsEMG measurements showed the EA-CV23 improving swallowing function in clinical practice. In the study, the EMG recording comparable to the PsEMG reflects the neuromuscular function, and the water consumption test mimicking only to some extent the WST to evaluate the ability of drinking water. (Discussion, Pages 14, Line 347-357; Methods, Pages 17-18 & 20-21, Line 428-433 & 520-521)

17. Page 3, line 62: "lick-swallow ratio" is not a standard measure of swallowing function in clinical practice – it's a measure used in animal studies only.

Response: We have made the revision accordingly. (Pages 3, Line 60-61)

18. Page 4, lines 70-73: "...pharyngeal surface electromyography (PsEMG, as an evaluation of the characteristics of external musculature by quantifying the root mean square, integrated electromyography and averaged electromyography) in the clinical practice..." What is meant by "external musculature"? What exactly is being measured clinically via EMG that translates to this animal study?

Response: This PsEMG method could reflect the underlying neuromuscular activity during swallowing movement. External musculature means the external surface muscle groups. In clinical practice, electrodes were attached to pharyngeal skin without inserting into the internal muscle with PsEMG. In that case, the signals collected by PsEMG might involve the activity of muscles underneath the surface skin attaching with the electrode. The electrode is likely to collect the responses from the superficial muscles, such as the digastric and mylohyoid, instead of that from the deeper muscles, including geniohyoid and hyoglossus. To target the mylohyoid connected to M1 and reduce the undesirable signals from other muscles as much as possible, , the electrode for EMG

recording in our study was inserted into the internal muscle presumed mylohyoid, which seems more suitable to reflect the neuromuscular activity of swallowing-related muscles (PMID: 17351080). (Pages 17-18, Lines 428-433)

19. Page 5, lines 97-100: “In the study, we found that optogenetic activation of the M1 L5 excitatory neurons could elicit frequency- and intensity-dependent EMG responses in mylohyoid, and unilateral photochemical ischemia of M1 was sufficient to produce dysphagia, establishing the PSD mouse model.” Separate these two “concepts” into different sentences for emphasis of importance. This is the first mention of optogenetics – explain why it was used here for clarification and added impact. Also briefly describe the PSD model with appropriate references – why was this model specifically chosen for this work?

Response: Thanks for your suggestion. The sentence has been separated into two sentences in the new version, and the emphasis has also been given on the optogenetics. (Pages 4, Line 90-92)

The reasons for choosing this model for this work have been expanded in the discussion part. Firstly, to investigate the role of M1 neurons in swallowing function, this model is an optimum choice for its specific damage in M1 brain region. Secondly, although there are kinds of used dysphagia animal models, such as transient middle cerebral artery occlusion (PMID: 31022397), Parkinson mouse model (PMID: 21459116, 26234713), and amyotrophic lateral sclerosis (PMID: 19107538, 31300881), the pathologies of these dysphagia models are much more complicated, which is not suitable for elucidating our purpose of the study. (Pages 12-13, Line 301-307)

20. Page 6, lines 115-116: “...the EGFP-positive neurons were detected in the M1 Layer 5 (L5) 96h instead of 48h or 72h after injection...” Please clarify why “instead of 48h or 72h after injection” is important information to know here. Why 48h and 72h, specifically?

Response: Considering that the infected animal could survive for 3~5d after PRV injection, most studies collected the PRV infected tissue at different timing including 24, 48, 72, and 96h (PMID: 32405000). Considering the life cycle of the PRV virus, it is speculated that the PRV could achieve 1~2 trans-synaptic tracing within 48h and 2~3 trans-synaptic tracing within 72h. Our results indicated that there might be more than 3 trans-synaptic tracing for PRV from mylohyoid to M1. These have been included in the revised text. (Pages 6, Line 111-113)

21. Page 6, lines 126-127: “Optogenetic activation of M1 neurons induced the increase in the mylohyoid activity and pharyngeal pressure...” How was pharyngeal pressure measured here? A brief summary would be helpful.

Response: Thank you for underlining this deficiency. The brief summary has been added to the text. The pressure was measured by a balloon which was filled with water and fixed on the pharyngeal near the root of tongue. (Pages 6, Line 124-126).

22. Pages 15-16, lines 386-387: "... and a tube (0.85*0.42 mm PE tube) for water delivery was placed in the mouse's hard palate." Please clarify "placed in" – this sounds like a surgical procedure. Do you mean "positioned against" rather than "placed in"?

Response: As you suggested, we have replaced the "placed in" with "positioned against". (Pages 18, Line 434)

23. Page 16, Line 438: "EA procedure in mice". Please clarify "EA" here in addition to using the acronym.

Response: This part has been clarified and supplemented in the revised text according to the comment. (Pages 20, Line 493)

24. Page 17, lines 442-443: "...at about a 0.5cm depth into CV23 acupoint, which located at the mylohyoid." Do you mean "at the presumed mylohyoid location" based on previous work (explain or cite here)? The details are included in the authors "rebuttal" but not incorporated into the manuscript.

Response: Thank you for pointing out this issue. More details of the location have been incorporated into the revised text. (Pages 20, Line 498-505)

25. Page 17, lines 438-445: "Electrical stimulation of 2 Hz with 1 mA strength of intermittent pulses for 15 min was performed...". Please clarify here why these parameters were specifically used. The details are included in the authors "rebuttal" but not incorporated into the manuscript.

Response: Thank you so much for the suggestions. More details have been included in the revised text to explain why the parameters were used here. (Pages 20, Line 504-514)

26. Page 17, lines 445-447: "... A sham EA control treatment followed the same procedure but used a non-electrical wooden toothpick instead of the stainless-steel needle...". Why not simply unplug the stainless-steel electrode from the stimulator for the control condition instead of using an entirely different electrode (different material, size, etc.) for the control condition?

Response: Thank you for the valuable comment. The method you mentioned is a good way to be adopted as sham EA procedure and has been accessible in the published studies (PMID: 32791039, 30348772). As for the method used in our study, it is also another way for considering the potentially unnecessary effect of piercing the needle into skin. Therefore, we choose the wooden toothpick for the control condition instead of unplugging the stainless-steel electrode from the stimulator. Both in clinical trials and basic animal studies (PMID: 24562381, 19433697, 30521869), this kind of way has also been used as a sham intervention. (Pages 20, Line 514-516)

27. Page 17, lines 449-456: Methodological details of the water swallow test are incomplete; therefore, replication efforts based on the available details would be impossible. What type of bottles and spouts were used, what volume of water did they hold, what type of water was used (pH-adjusted filtered tap water or DI water), was the water room temperature, etc. Did the mice drink in their home cage or were they tested individually in separate cages? (Also clarify if mice were single housed for the entire experiment – this detail is missing). Were the bottles leak-tested to determine the amount of water that spontaneously leaks from the bottles during the typical water swallow test duration (without a mouse drinking from the spout)?

Response: We are grateful for this suggestion. To be more clear and in accordance with your concerns, we have added these methodological details as follows: The sipper tube bottles were constructed with a 50mL centrifuge tube and a silicone stopper. Then a metal spout with a stainless-steel ball and a stainless-steel pressure bar were connected with a leakproof sealing ring to prevent the water from leaking. Each mouse was fed individually in the cage during test. The water in the bottle (pH, ~7.4; hardness, 108 mg/L; chromaticity, less than 5.) was filtered from the tap water. We have not found the water spontaneously leaking from the bottles. (Pages 20-21, Line 520-531)

Reviewer #2 (Remarks to the Author):

I am broadly satisfied with the way in which the authors have addressed my concerns as a reviewer but note that other reviewers have raised more specialized concerns about the techniques used I am not able to evaluate. However as far as the concerns I raised about the paper I am satisfied for it to be published.

Response: Thank you very much for your satisfaction and recommendation.

Reviewer #3 (Remarks to the Author):

1. For c-Fos induction shown in Sup. Figure 6: there is no control Fos with EA.

Response: Thank you for pointing out this issue. The results of control c-Fos with EA have been added in the revised text as **Sup.Fig.6**. The new results showed that c-Fos expression in the mentioned brain regions except for LSV (Ventral part of the lateral septal nucleus) showed increased c-Fos expression responding to EA-CV23 stimulation compared to control treatment (data of LSV shown below), indicating the activation of neurons in LSV was not specific to EA-CV23 stimulation.

Representative images showing c-Fos expression in LSV after control or EA-CV23 treatment (a); and the statistical analysis was plotted (b). Scale bar, 100 μ m.

2. A new issue emerges. The whole studies tried to suggest information flow from M1 to PBN and finally to NTS in driving swallowing. However, the tools used appeared not to rule out a reverse direction: NTS-PBN-M1. Could AAV2/9 injection in M1 retrogradely label PBN neurons projecting to M1, and could the reported monosynaptic connections between M1-PBN be due to antidromic activation of PBN terminals in M1 and back to soma? AAV2/1 anterograde labeling is known to contain some retrograde activity. As such AAV2/1 injection in M1 may also retrogradely label PBN neurons. AAV2/1 injection in PBN could also retrogradely label NTS neurons. The strongest evidence for M1-PBN-NTS direction is the silencing of NTS firing in response to M1 stimulation following PBN silencing, although some forms of reciprocal connections between M1-projecting PBN neurons and NTS neurons may need to be ruled out. The authors may need to address some of these concerns and then provide some discussions if they cannot fully address this possibility.

Response: Thank you for pointing out these concerns. To address your issues, we have provided more direct evidence on demonstrating the function of M1-PBN-NTS neural circuit (**Fig.6j-k**), and structural connections (**Sup.Fig.7** or represented below). These results further confirmed the existence of M1-PBN-NTS neural circuit, while the possibility of the NTS-PBN-M1 neural circuit has not been ruled out. Thus, the NTS-PBN-M1 direction neural circuit you concerned is possible that might involve the information ascending from the mylohyoid to the M1. To better address the problems you concerned, we listed your questions and answered them as follows.

(1) For the issue ‘Could AAV2/9 injection in M1 retrogradely label PBN neurons projecting to

M1'?

Response: This question is very good. As you mentioned, it is happened that AAV2/9 virus could trace retrogradely or anterogradely, which learned from the discussions with the colleagues. It is worth noting that it is rarely occurred that the virus traced retrogradely or anterogradely when the titer of virus was at 10^{12} used in the study. To confirm this result and exclude this retrograde labeling for the virus, we observed only the terminals but no soma in PBN (see below), and this indicated that there was no retrogradely label for this AAV2/9. Meanwhile, the optogenetics mostly used AAV2/9 as vector, and nearly no evidence reported this AAV2/9 could retrogradely label the neurons. Therefore, the effect we observed was not likely caused by this retrograde labeling for AAV2/9.

Images showing AAV2/9-CaMKII α -mCherry virus injected into M1, while only terminals in PBN were labeled and no soma was observed. Scale bar, 50 μ m.

(2) For the issue ‘could the reported monosynaptic connections between M1-PBN be due to antidromic activation of PBN terminals in M1 and back to soma’?

Response: Thanks for your nice question. As you are concerned, the optogenetic activation of the terminal could cause ions back to soma and in turn induce the antidromic action potential firing, and this phenomenon would indeed contaminate the results. There is no doubt that this is the technical issue in the field, which needs to be broken through in the future. In the study, to rule out this possibility, we injected AAV2/1-Cre virus into M1 and AAV2/9-Dio virus injected into PBN, and soma in PBN was observed, supporting the existence of M1-PBN circuit (**Fig.5g**). It is true that AAV2/1 also could retrogradely trace and could not answer this question perfectly. Other results showed that neurons in PBN labeled when anterograde VSV was injected into M1 (**Fig.5h**), and that neurons in M1 labeled when CTB-555 was injected into PBN (**Sup. Fig.7**). Besides, chemoinhibiting of PBN neurons decreased the mylohyoid activity induced by optogenetic activation of M1 (**Fig.5d**, AAV2/9-CaMKII α -ChR2-mCherry was injected into M1 and light stimulation, while AAV2/9-CaMKII α -hM4Di-mCherry was injected into PBN and CNO intraperitoneal injection). Therefore, the effects we observed might be not likely caused by the antidromic activation.

(3) The strongest evidence for M1-PBN-NTS direction is the silencing of NTS firing in response to M1 stimulation following PBN silencing, although some forms of reciprocal connections between M1-projecting PBN neurons and NTS neurons may need to be ruled out.

Response: We agree with your comments. To validate the M1-PBN-NTS neural circuit, *in vivo* electrophysiological recording results showed that the neural activity in NTS was significantly decreased by chemoinhibiting PBN neurons in response to M1 activation (M1⁺+PBN⁻, **Fig.6e-f**), and the neural activity in NTS decreased after selectively chemoinhibition of PBN neurons innervated by M1 (M1^{Cre}-PBN^{Dio-hM4Di}-NTS^{Electrode}, **Fig.6j-k**). Moreover, according to your suggestions, we have injected CTB-555 or CTB-488 into PBN and M1, respectively, and observed the fluorescent neurons in NTS or PBN (see below). This preliminary result suggested the possibility of NTS-PBN-M1 direction neural circuit. Although the possibility of reverse NTS-PBN-M1 direction neural circuit has not been ruled out, the above results (**Fig.6e-f, j-k**) provided the evidence suggesting the function of M1-PBN-NTS direction neural circuit. We agree with you that there might be NTS-PBN-M1 direction neural circuit except for the M1-PBN-NTS neural circuit, while whether this circuit could participate in transmitting the information from NTS to M1 needs further investigation. We are trying to dig out how the M1, PBN and NTS interact with others and modulate swallowing process during EA-CV23 stimulation.

We have added the discussions into the revised text. (Pages 13-14, Line 317-341)

CTB-555 injected into PBN and the fluorescent neurons were detected in NTS (left), while CTB-488 injected into M1 and the neurons in PBN were labeled. Scale bar, 50 μ m.

REVIEWER COMMENTS

Reviewer #1 (Remarks to the Author):

Although the authors report they have “carefully scrutinized the entire manuscript and asked a native and professional English language speaker” for assistance, the entire manuscript (i.e., nearly every sentence, heading/subheading, and figure title/legend) still requires extensive English language editing before any acceptance decision can be made. Currently, the manuscript remains quite challenging to read through and fully understand. Aside from this major language-based concern, the authors have sufficiently addressed the majority of the scientific-based comments/concerns raised during the review process. A few methodological and animal welfare uncertainties remain, as listed below:

1. Details of how the EMG wire electrodes were inserted into the mylohyoid muscle in awake mice remain incomplete.
 - a. How exactly was the awake mouse “fixed to a mouse adapter in the supine position” (Page 18, Line 434) during EMG wire insertion and subsequent EMG recording? How was struggling/escape/stress behavior prevented?
 - b. How were the EMG electrodes inserted into the mylohyoid muscle (Page 18, Lines 435-436)? For example, via needle (what size?) insertion through the skin? How was this procedure completed without causing pain/discomfort in awake mice?
 - c. Also, was a single pair of EMG wires inserted unilaterally into the mylohyoid (which side?)? Or was a pair of EMG wires inserted on each side of the mylohyoid for bilateral EMG recording?

2. Details of the “balloon pressure detection experiment” are improved but remain incomplete.
 - a. Only in Figure 1f schematic is the term “pharyngeal pressure transducer” used – include this wording in the methods section as well for added clarity.
 - b. Also, the wording “...measure the pressure of the balloon when mice began to drink” (Page 19, Lines 464-465) is misleading, as it gives the impression that the mice were freely drinking when they were not. Please clarify this wording for accuracy. Were there any instances of water running from the nose during water delivery? This finding would suggest the water delivery rate was too high for the mice to handle.
 - c. Please clarify the updated wording on Page 6 (Lines 125-126): “...pressure was measured by the balloon filled with water and fixed on the pharyngeal (missing a word here) near the root of the tongue.” Describe how it was “fixed” – For example, glued/cemented, sutured, or simply held in place via the tongue?

Reviewer #3 (Remarks to the Author):

The authors have carefully addressed my concerns, and the revised discussion has highlighted the possible involvement of the reverse NTS-PBN-M1 circuits.

based on the writing in addressing my comments (pages 13-14), some additional English editing is needed.

Point-by-point responses (MS ID: NCOMMS-21-36223B)

Reviewer #1 (Remarks to the Author):

Although the authors report they have “carefully scrutinized the entire manuscript and asked a native and professional English language speaker” for assistance, the entire manuscript (i.e., nearly every sentence, heading/subheading, and figure title/legend) still requires extensive English language editing before any acceptance decision can be made. Currently, the manuscript remains quite challenging to read through and fully understand. Aside from this major language-based concern, the authors have sufficiently addressed the majority of the scientific-based comments/concerns raised during the review process. A few methodological and animal welfare uncertainties remain, as listed below:

Response: Thanks for your suggestions. To improve the English language and make the manuscript read easily, English language has been extensively reedited. More details of the methodological and animal welfare have also been provided.

1. Details of how the EMG wire electrodes were inserted into the mylohyoid muscle in awake mice remain incomplete.

a. How exactly was the awake mouse “fixed to a mouse adapter in the supine position” (Page 18, Line 434) during EMG wire insertion and subsequent EMG recording? How was struggling/escape/stress behavior prevented?

Response: Thanks for you pointing this issue. Mice were first anesthetized with isoflurane before this fixation and electrode insertion. To reduce their struggling/stress, these experimental procedures should be finished as soon as possible. To avoid the obvious struggle/escape/stress of mice during the whole recording, the anesthetic mask connected with isoflurane was available to the mouse nostrils. These details have been included into the new revision. (Page 19, Lines 445-450)

b. How were the EMG electrodes inserted into the mylohyoid muscle (Page 18, Lines 435-436)? For example, via needle (what size?) insertion through the skin? How was this procedure completed without causing pain/discomfort in awake mice?

Response: The electrodes (about 15mm diameter) were inserted into the presumed mylohyoid through the skin. Before this manipulation, mice were first anesthetized with isoflurane to reduce pain or discomfort. (Page 19, Lines 447-452)

c. Also, was a single pair of EMG wires inserted unilaterally into the mylohyoid (which side)? Or was a pair of EMG wires inserted on each side of the mylohyoid for bilateral EMG recording?

Response: Thanks for you pointing this concern. A pair of EMG electrodes was inserted unilaterally into the mylohyoid in the left side, and this important detail has been supplemented into the new revision. (Page 19, Lines 447-449)

2. Details of the “balloon pressure detection experiment” are improved but remain incomplete.

a. Only in Figure 1f schematic is the term “pharyngeal pressure transducer” used – include this

wording in the methods section as well for added clarity.

Response: Thanks for your nice suggestion. The “pharyngeal pressure transducer” shown in Figure. 1f is also called as a baroreceptor mentioned in the Methods section before, and this has been included into Methods section in the new revision accordingly. (Page 20, Lines 478-479)

b. Also, the wording “...measure the pressure of the balloon when mice began to drink” (Page 19, Lines 464-465) is misleading, as it gives the impression that the mice were freely drinking when they were not. Please clarify this wording for accuracy. Were there any instances of water running from the nose during water delivery? This finding would suggest the water delivery rate was too high for the mice to handle.

Response: Sorry for this misleading writing. Here, in this Balloon pressure detection experiment, instead of data of water-induced pressure of balloon, the data of blue light stimulation-induced pressure have shown in this paper (Fig. 1d-g). This mistake has been corrected accordingly. (Page 20, Lines 482-486)

As for your concerns about the water running from the nose, we have added more details of water delivery. To avoid water running from the nose, mice were in a semireclined position, and water was delivered at least 1 min intervals to ensure mice had enough time to swallow the water. (Page 19, Lines 462-464)

c. Please clarify the updated wording on Page 6 (Lines 125-126): “...pressure was measured by the balloon filled with water and fixed on the pharyngeal (missing a word here) near the root of the tongue.” Describe how it was “fixed” – For example, glued/cemented, sutured, or simply held in place via the tongue?

Response: Thanks for you pointing this out. We have made the revision as you suggested accordingly. “...the balloon filled with water and held in the pharyngeal position near the root of tongue”. (Page 6, Lines 128-129)

Reviewer #3 (Remarks to the Author):

The authors have carefully addressed my concerns, and the revised discussion has highlighted the possible involvement of the reverse NTS-PBN-M1 circuits.

based on the writing in addressing my comments (pages 13-14), some additional English editing is needed.

Response: Thanks for your suggestion. The writing has been edited. (Page 14-15, Lines 327-351).